# Supervised Fine-Tuning or Contrastive Learning? Towards Better Multimodal LLM Reranking

**Xin Zhang**[1,2]*, **Ziqi Dai**[1]*, **Mingxin Li, Yanzhao Zhang, Dingkun Long**
**Pengjun Xie, Meishan Zhang**[1]†, **Wenjie Li**[2], **Min Zhang**[1]
[1]Harbin Institute of Technology, Shenzhen    [2]The Hong Kong Polytechnic University
{ziqi.dai,zhangxin2023}@stu.hit.edu.cn
Model released at https://hf.co/vec-ai/lychee-rerank-mm

## Abstract

In information retrieval, training reranking models focus mainly on two types of objectives: metric learning (*e.g.,* contrastive loss to increase predicted scores on relevant query-document pairs) and classification (binary label prediction of relevance vs. irrelevance). For BERT-style encoders, various studies have shown that contrastive learning (CL) can be more effective than discriminative (classification) learning. However, for large language models (LLMs), classification via supervised fine-tuning (SFT), which predicts "yes" (*resp.* "no") token for relevant (*resp.* irrelevant) pairs, appears more promising as it aligns well with the generative nature of LLMs. This divergence raises a central question: *which objective is intrinsically better suited to LLM-based reranking, and what mechanism underlies the difference?* In this work, we conduct a comprehensive comparison and analysis between CL and SFT for reranking, taking the universal multimodal retrieval (UMR) as the experimental playground. We first decompose the objectives into two components: *weight*, which controls the magnitude of those updates, and *direction*, which guides the model updates, then present a unified framework for understanding their interactions. Through probing experiments, we find that SFT provides a substantially stronger weighting scheme than CL, whereas the preferred scoring direction shows no clear winner. Taken together, these results point to a consistent advantage of SFT over CL for LLM reranking. To further validate our findings, we conduct large-scale training with SFT and present new state-of-the-art rerankers on the compiled MRB benchmark. We also provide ablations on SFT settings and expect our findings to benefit future research and applications.

## 1 Introduction

Reranking is a crucial step in the retrieval pipeline (Lin et al., 2022), aiming to refine the initial results obtained from the previous search stage by reordering them based on their relevance to a given query. In recent years, the integration of Large Language Models (LLMs) into reranking techniques has shown promising results in text retrieval (Ma et al., 2024b) and has gradually become the standard approach (Sharifymoghaddam et al., 2025). When extending to the multimodal setting (Liu et al., 2023; Wei et al., 2025), multimodal LLMs (MLLMs) also become the promising backbone choice (Lin et al., 2025; Zhang et al., 2025b) as their strong multimodal understanding capabilities.

Current widely used rerankers are typically in the point-wise setting (Lin et al., 2022), which independently scores each query-candidate pair and ranks the candidates. The simple architecture of point-wise rerankers makes them easy and efficient to applicate in real-world scenarios, and there emerges various open-source state-of-the-art (SOTA) models (Chen et al., 2024; Zhang et al., 2024), particularly LLM-based ones (Sharifymoghaddam et al., 2025; Zhang et al., 2025d). To train such rerankers[1], one straightforward approach follows the pre-LLM practice of contrastive learning (CL)

---

*Equal contribution, order determined by random dice toss. †Corresponding author: Meishan Zhang.
[1]Throughout this work, reranking primarily refers to point-wise reranking setting.

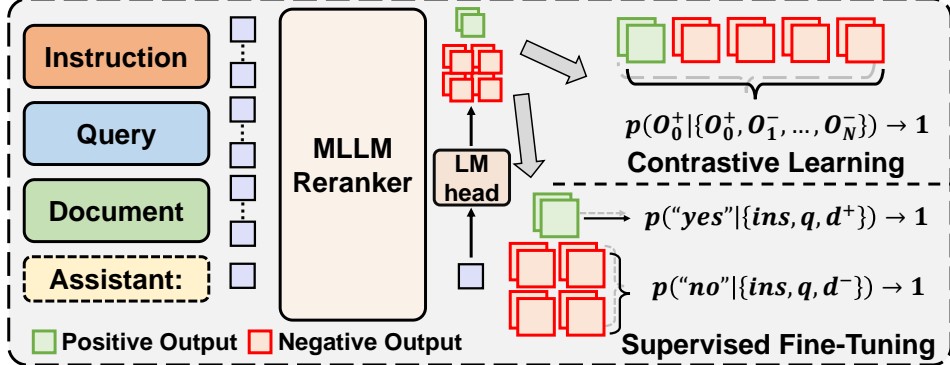

Figure 1: Comparison of Supervised Fine-Tuning (SFT) and Contrastive Learning (CL) for the MLLM reranker. SFT treats reranking as a binary classification task ("yes"/"no"). In contrast, CL trains the model to directly distinguish the positive document from a list of negatives.

(Nogueira et al., 2019; Zhang et al., 2024), computing InfoNCE loss (Oord et al., 2018) one predicted relevance scores. Another approach is to directly perform supervised fine-tuning (SFT) (Nogueira et al., 2020; Zhang et al., 2025d), which optimizes the model to predict the next token ("true/yes" for relevant, "false/no" for irrelevant) and takes the "true/yes" token probability as the relevance score. The illustration of them is shown in Figure 1. Before the emergence of LLMs, contrastive learning was the dominant approach for leveraging BERT-style encoders due to its strong performance (Nogueira et al., 2019; Zhang et al., 2024). However, SFT are now widely applied to LLMs (Nogueira et al., 2020; Zhang et al., 2025d) and appears to deliver competitive results. This raises a natural research question: *which objective is intrinsically better for LLM reranking, and why?*

Meanwhile, research on multimodal reranking remains largely restricted to single datasets or narrowly defined tasks (Xu et al., 2025), limiting the generalizability of existing approaches. Building on recent advances in universal multimodal retrieval (Zhang et al., 2025b), our objective is to develop a universal multimodal reranking model that can consistently adapt across diverse modalities.

In this work, we aim to explore the question by providing an analysis and empirical comparison of the two approaches on the universal multimodal retrieval task as testbed. We first design the General Multimodal Reranker (GMR, §3.1), and then analyze the two training approaches and decompose their loss functions (§3.3) into *weight* and *direction*. Based on this, we implement a unified framework for CL and SFT losses and conduct experiments to compare and analysis them (§4). To make comprehensive evaluations of multimodal reranking, we compile a new unified benchmark called *MRB* (multimodal reranking benchmark, §5).

Through analysis and comparison, we find that SFT consistently outperforms CL for LLM-based rerankers, and: (1) The weight component, rather than the direction, accounts for the most performance gap; (2) A larger weight improves robustness to numerical errors in training, where SFT intrinsically assigns larger weights than CL; (3) The function of weight is a input-specific guidance: down-weight already-well-learned input pairs and up-weight hard or under-fit pairs; (4) The native SFT direction is almost optimal, whereas CL can be further improved by tuning its direction matrix. To further validate the potential of SFT, we train two reranking models (*i.e.,* GMR-3B and GMR-7B), which set new state-of-the-art results on MRB. We will release code, data and models to facilitate future research in this area. Our contributions are:

- We provide a unified analysis of SFT and CL for LLM-based reranking, showing that SFT intrinsically outperforms CL. By decomposing the loss into *weight* and *direction* components, we reveal that SFT's weight term delivers stronger optimization signals.

- We introduce the MRB benchmark, comprising 40 datasets across single-, cross-, and fused-modal retrieval, offering a comprehensive evaluation for universal multimodal reranking.

- We develop GMR models, instruction-aware multimodal LLM rerankers trained on 1.5M diverse pairs. GMR-3B and GMR-7B achieve state-of-the-art results on MRB, highlighting the effectiveness of SFT and providing strong backbones for future research.

## 2 RELATED WORK

**Reranking with Large Language Model.**  Reranking improves retrieval output quality by jointly modeling the query and retrieved candidates and reorder the candidates (Lin et al., 2022). In recent years, reranking is dominated by methods based on pretrained language models (Nogueira et al., 2019; 2020), with LLM-based approaches becoming particularly prominent in the latest advancements (Ma et al., 2024b; Zhuang et al., 2024; Sharifymoghaddam et al., 2025; Zhang et al., 2025c). Compared to the widely studied list-wise reranking (Ren et al., 2025; Liu et al., 2025), in this work, we focus on the more straightforward and widely used *point-wise* approach (Zhang et al., 2024; Guo et al., 2025), which scores each query-candidate pair independently and ranks the candidates.

Training point-wise rerankers has traditionally relied on contrastive learning (CL) (Nogueira et al., 2019; Zhang et al., 2024), which is also a verified choice for LLM-based models (Ma et al., 2024b). However, for such generative language models, a supervised fine-tuning (SFT) approach (Nogueira et al., 2020) seems to be more aligned with the model nature, as it directly optimizes the model to predict the next token ("true/yes" for relevant, "false/no" for irrelevant) based on the input query and candidate, rather than relying on a contrastive loss that compares the relevant and irrelevant candidates. There is no clear consensus on which approach is better yet. To bridge this significant research gap, we conduct an analysis with empirical comparison of the two approaches, and demonstrate that SFT outperforms CL in terms of performance.

**Multimodal Information Retrieval.**  Multimodal Retrieval aims to retrieve relevant candidates from and based on modalities beyond text (Wang et al., 2024), which involves various sub-tasks such as image-text retrieval (Cao et al., 2022), composed image retrieval (Song et al., 2025), and text-image interleaved retrieval (Zhang et al., 2025a). Recent advancements in this field have been shifted to a more generalized view, exploring the universal multimodal retrieval (UMR) (Liu et al., 2023; Wei et al., 2025; Zhang et al., 2025b) which compile a wide range of datasets and tasks into a unified benchmark. Retrievers (Lin et al., 2025; Zhang et al., 2025b) driven by multimodal LLMs have shown significant improvements in understanding and processing multimodal data, enabling more effective retrieval across different modalities. While the reranking stage is crucial for enhancing the precision of retrieval system, it has been less studied in UMR (Lin et al., 2025). In this work, we investigate how to build better LLM reranking models, presenting state-of-the-art MLLM-based rerankers for UMR.

## 3 METHOD

In this work we analyze the contrastive learning (CL) and supervised fine-tuning (SFT) approaches to reranking, taking the multimodal retrieval as the experimental playground. We first introduce our reranking model (§3.1), training by CL or SFT (§3.2), and then present our tools for analysis (§3.3).

### 3.1 RERANKER IMPLEMENTATION

Our general multimodal reranker (namely GMR) follows the conventional design of LLM-based point-wise reranking models. We employ a strong MLLM as the backbone, which could process diverse input modalities, encompassing images, text, and multimodal combinations.

**Instruction-Aware Reranking.**  Given query $q$ and document $d$, we set an instruction $ins.$ to describe detailed task objectives, which has proven highly effective in MLLM-based multimodal retrieval (Lin et al., 2025; Zhang et al., 2025b). For example, in the Visual Document Retrieval (Ma et al., 2024a; Faysse et al., 2025), we use an instruction "*Find a screenshot that relevant to the user's question.*" to guide the model to better evaluate the relevance between query and visual document. We list all instructions of our model in Appendix D.3. The inputs are in the form of $(ins., q, d)$ and formatted into the template shown in Figure 8 before being fed into the MLLM backbone.

**Relevance Score Computation.**  In the **SFT** setting, given the task instruction $ins.$, query $q$, and document $d$, our reranker computes the LM-head logits of the tokens "yes" and "no", and uses the normalized probability of the "yes" logit as the similarity score $s$. This process could be formally

expressed as:

$$s(ins., q, d) = \frac{\exp(z^y)}{\exp(z^y) + \exp(z^n)}, \tag{1}$$

where $z^y$ and $z^n$ denote the LM-head logits corresponding to the tokens "yes" and "no", respectively, given the instruction, query, and document as context. With such relevance scores, we could rerank all retrieved candidates more precisely. This method is more aligned to the generative nature of MLLM and thus allows us to leverage its powerful understanding ability while providing an effective scoring mechanism for reranking purposes.

In the **CL** setting, the relevance score is the LM-head logit of the token "yes" only:

$$s(ins., q, d) = z^y. \tag{2}$$

## 3.2 RERANKER TRAINING

In the training of reranking, each data example contains one query $q$, one relevant document (positive) $d_0^+$, and several irrelevant documents (negatives, the selection is described in Appendix C.3) $\{d_1^-, d_2^-, \ldots, d_N^-\}$. As shown in Figure 1, we explore both CL and SFT based training.

• **Contrastive Learning:** With relevance score $s$ from Equation 2, we compute the InfoNCE loss (Oord et al., 2018) for each example:

$$\mathcal{L}^{\mathrm{CL}} = -\log \frac{\exp(s(ins., q, d_0^+))}{\exp(s(ins., q, d_0^+)) + \sum_i \exp(s(ins., q, d_i^-))}. \tag{3}$$

• **Supervised Fine-Tuning:** The objective is independently predicting correct label ("yes" or "no") for each input pair. We reorganize each example into multiple triplets $(ins., q, d_i)$, with each triplet corresponding to a distinct document $d_i$. For each triplet, the model predicts the LM-head logits of "yes" and "no", then we compute the cross-entropy loss using the ground-truth label $l$:

$$\mathcal{L}_i^{\mathrm{SFT}} = -\log(p(l|z(\{\text{"yes"},\text{"no"}\}|\{ins., q, d_i\}))), \tag{4}$$

where $z(\{\text{"yes"},\text{"no"}\}|\{ins., q, d\})$ denotes the LM-head logits of "yes" and "no". The relevance label $l$ is "yes" for positive documents and "no" for negatives. This loss encourages the model to assign higher probabilities to the correct label, thereby improving the ranking performance.

## 3.3 LOSS FUNCTION DECOMPOSITION

We analyze two reranking loss functions by decomposing them into two key components: *weight* and *direction*. With this decomposition, we perform probing experiments in §4.

**Basic Notation.** We denote the SFT-style data instance with positive (*resp.* negative) doc as $o_0^+ = \{ins., q, d_0^+\}$ (*resp.* $o_i^- = \{ins., q, d_i^-\}, i = 1, 2, \ldots, N$). The reranker is conceptualized as two components: a mapping function $f(\cdot|\theta)$ (parameterized by $\theta$) that converts $o_i$ to the feature representation $\mathbf{h}_i = f(o_i|\theta)$, and a transformation $\mathcal{M}^y$ that maps $\mathbf{h}_i$ into the LM-head logit of "yes" token $z^y(h_i) = \mathbf{h}_i \cdot \mathcal{M}^y$, while the LM-head logit of "no" token $z^n$ could be computed by $\mathcal{M}^n$.

**Unified View.** From §3.2, the SFT loss is calculated separately for each positive or negative doc of an example, while the CL loss is computed in an integrated manner across all positive and negative docs of the same example. To enable a fair comparison, we adopt the total loss $\mathcal{L}(\{o_i\}_{i=0}^N, \theta)$ over an entire example (with one positive and $N$ negatives) as the unit of analysis. So we have the gradient

$$\frac{\partial \mathcal{L}}{\partial \theta} = \frac{\partial \mathcal{L}}{\partial \mathbf{h}_0^+} \frac{\partial \mathbf{h}_0^+}{\partial \theta} + \sum_i \frac{\partial \mathcal{L}}{\partial \mathbf{h}_i^-} \frac{\partial \mathbf{h}_i^-}{\partial \theta}, \tag{5}$$

where $\mathbf{h}_0^+$ is the feature of positive doc and $\mathbf{h}_i^-$ is that of $i$-th negative.

To understand the influence of positive and negatives on the model, we calculate the partial derivative of the loss function with respect to the hidden state. For CL, we only use "yes" token, and by substituting the specific loss (Equation 3) into the gradient, we obtain the partial derivatives:

$$-\frac{\partial \mathcal{L}^{\mathrm{CL}}}{\partial \mathbf{h}_0^+} = \frac{\sum_j \exp(z^y(\mathbf{h}_j^-))}{\exp(z^y(\mathbf{h}_0^+)) + \sum_i \exp(z^y(\mathbf{h}_i^-))} \mathcal{M}^y, \tag{6}$$

$$-\frac{\partial \mathcal{L}^{\mathrm{CL}}}{\partial \mathbf{h}_i^-} = -\frac{\exp(z^y(\mathbf{h}_i^-))}{\exp(z^y(\mathbf{h}_0^+)) + \sum_i \exp(z^y(\mathbf{h}_i^-))}\mathcal{M}^y. \tag{7}$$

In SFT, we first merge the Equation 4 of multiple pairs in one example into the total loss

$$\mathcal{L}^{\mathrm{SFT}} = -\log\frac{\exp(z^y(\mathbf{h}_0^+))}{\exp(z^y(\mathbf{h}_0^+)) + \exp(z^n(\mathbf{h}_0^+))} - \sum_i \log\frac{\exp(z^n(\mathbf{h}_i^-))}{\exp(z^y(\mathbf{h}_i^-)) + \exp(z^n(\mathbf{h}_i^-))}.$$

Then we have partial derivatives

$$-\frac{\partial \mathcal{L}^{\mathrm{SFT}}}{\partial \mathbf{h}_0^+} = \frac{\exp(z^n(\mathbf{h}_0^+))}{\exp(z^y(\mathbf{h}_0^+)) + \exp(z^n(\mathbf{h}_0^+))}(\mathcal{M}_y - \mathcal{M}_n), \tag{8}$$

$$-\frac{\partial \mathcal{L}^{\mathrm{SFT}}}{\partial \mathbf{h}_i^-} = -\frac{\exp(z^y(\mathbf{h}_i^-))}{\exp(z^y(\mathbf{h}_i^-)) + \exp(z^n(\mathbf{h}_i^-))}(\mathcal{M}_y - \mathcal{M}_n). \tag{9}$$

The complete derivation of the above process is provided in the Appendix A.2.

**Loss Decomposition.** As above gradients looks similar, we can break them down into two parts: *weight* and *direction*. They reflect the differences between CL and SFT.

• *Weight $W$* is a scalar that controls the magnitude of the updates. From Equation 6 - 9, we obtain the weights as shown below:

$$W_{\mathrm{CL}}^+ = \frac{\sum_i \exp(z^y(\mathbf{h}_i^-))}{\exp(z^y(\mathbf{h}_0^+)) + \sum_i \exp(z^y(\mathbf{h}_i^-))}, \tag{10}$$

$$W_{\mathrm{CL}}^- = \frac{\exp(z^y(\mathbf{h}_i^-))}{\exp(z^y(\mathbf{h}_0^+)) + \sum_i \exp(z^y(\mathbf{h}_i^-))}, \tag{11}$$

$$W_{\mathrm{SFT}}^+ = \frac{\exp(z^n(\mathbf{h}_0^+))}{\exp(z^y(\mathbf{h}_0^+)) + \exp(z^n(\mathbf{h}_0^+))}, \tag{12}$$

$$W_{\mathrm{SFT}}^- = \frac{\exp(z^y(\mathbf{h}_i^-))}{\exp(z^y(\mathbf{h}_i^-)) + \exp(z^n(\mathbf{h}_i^-))}. \tag{13}$$

Compared with CL, $W_{SFT}$ only focus on the single document, without the interactions with all negatives of the same query like CL.

• *Direction $D$* is a vector that controls the direction of model updates. From Equation 6 and 8, the direction from the positive $d^+$ for CL is $D_{\mathrm{CL}}^+ = \mathcal{M}_y$, and that of SFT is $D_{\mathrm{SFT}}^+ = \mathcal{M}_y - \mathcal{M}_n$. While from Equation 7 and 9, direction from negatives are $D_{\mathrm{CL}}^- = -\mathcal{M}_y$ and $D_{\mathrm{SFT}}^- = -(\mathcal{M}_y - \mathcal{M}_n)$. Apparently, for both CL and SFT, the update directions of positive and negatives are opposite.

---

**Algorithm 1** Unified Reranking Loss

**Require:** inputs $\mathcal{O} \leftarrow \{o_0^+, \ldots, o_n^-\}$
**Ensure:** loss value $\mathcal{L}$
1: $\mathcal{M} \leftarrow \mathtt{lm\_head}(\text{"yes"},\text{"no"})$
2: $logits \leftarrow \mathcal{M} \cdot f(\mathcal{O}|\theta)$
 *//— weight branch ————————-*
3: **if** $\mathtt{weight}=\text{"sft"}$ **then**
4: $\quad s \leftarrow \mathrm{Softmax}(logits)[0].detach()$
5: $\quad W^+ \leftarrow W_{\mathrm{sft}}^+ \leftarrow 1 - s[0]$
6: $\quad W^- \leftarrow W_{\mathrm{sft}}^- \leftarrow s[1:]$
7: **else** $\qquad\qquad\qquad \triangleright \mathtt{weight}=\text{"cl"}$
8: $\quad s \leftarrow \mathrm{Softmax}(logits[0]).detach()$
9: $\quad W^+ \leftarrow W_{\mathrm{cl}}^+ \leftarrow 1 - s[0]$
10: $\quad W^- \leftarrow W_{\mathrm{cl}}^- \leftarrow s[1:]$
11: **end if**
 *//— direction branch ——————-*
12: $M_y \leftarrow logits[:,0]; \quad M_n \leftarrow logits[:,1]$
13: **if** $\mathtt{direction}=\text{"sft"}$ **then**
14: $\quad D^+ \leftarrow D_{\mathrm{sft}}^+ \leftarrow M_n[0] - M_y[0]$
15: $\quad D^- \leftarrow D_{\mathrm{sft}}^- \leftarrow M_y[1:] - M_n[1:]$
16: **else** $\qquad\qquad \triangleright \mathtt{direction}=\text{"cl"}$
17: $\quad D^+ \leftarrow D_{\mathrm{cl}}^+ \leftarrow -M_y[0]$
18: $\quad D^- \leftarrow D_{\mathrm{cl}}^- \leftarrow M_y[1:]$
19: **end if**
 *//————————————————-*
20: $\mathcal{L} \leftarrow \mathrm{mean}(W^+ D^+ + \sum_i W_i^- D_i^-)$
21: **return** $\mathcal{L}$

---

In summary, CL and SFT share similar direction components, and we believe that differing initializations[2] are insufficient to account for performance differences. In contrast, *CL computes the weight using all positive and negative documents within a sample, while SFT assigns weights independently per document*, making this the likely key factor in performance variation.

**Unified Framework**   Building on the above decomposition, we propose a unified reranking loss framework (URL), with pseudo-code provided in Algorithm 1. This framework allows us to independently analyze *weight* and *direction*, thereby facilitating a deeper understanding of the differences between the two training paradigms through controlled adjustments during computation. We then validate our analysis through probing experiments in the following §4.

---

[2]$\mathcal{M}_y$ compared to $\mathcal{M}_y - \mathcal{M}_n$.

## 4 ANALYSIS

In this section, we continue and validate the analysis of §3.3 through probing experiments. We choose universal multimodal retrieval as the testbed, compiling a new benchmark (MRB §5.1) includes single-modal tasks (text-to-text, image-to-image), cross-modal tasks (*e.g.,* text-to-image), as well as fused-modal tasks (either the query or the document could consist of text + image). We defer the description of experiment settings and evaluation benchmark to §5.1.

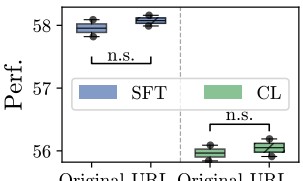

**General Empirical Comparison.** We first train both CL and SFT rerankers with the original implementation and our URL framework to (1) find the winner in practice, and (2) verify that URL faithfully reproduces the original implementation, supporting the subsequent analyses built on URL. As shown in Figure 2, under the identical setting, *SFT consistently outperforms CL.* Meanwhile, URL yields statistically indistinguishable performance to the original. It thus could be trusted in the following analysis.

Figure 2: Performance comparison of the original implementations and our URL.

**Weight $W$ Dominates Performance.** To investigate why SFT outperforms CL, we first dissect the contribution of weight and direction. In Table 1, we train the model with all combinations by URL. We observe that the improvements from weight (*i.e.,* $\Delta_W$) is more significant than that of direction ($\Delta_D$). This suggests that the weight $W$ is the dominant factor in the performance gap between SFT and CL, guiding us to focus on the weight in the following section. However, the direction also contributes to the gap, which is investigated in §4.2.

|  | $D_{\text{SFT}}$ | $D_{\text{CL}}$ | $\Delta_D$ |
|---|---|---|---|
| $W_{\text{SFT}}$ | 58.09 | 57.88 | ▼ 0.21 |
| $W_{\text{CL}}$ | 56.99 | 56.40 | ▼ 0.59 |
| $\Delta_W$ | ▼ **1.10** | ▼ **1.48** | |

Table 1: MRB results of all loss components combinations, where the weight $W$ delivers the dominant influence on performance.

### 4.1 FUNCTION OF WEIGHT

To figure out why $W_{\text{CL}}$ is less effective that $W_{\text{SFT}}$ and what is the function of $W$, we start from the observation of (Chen et al., 2021). In small-batch CL training with InfoNCE, *gradients would shrink to very small scale, close to random precision errors, and thus cease to provide effective learning guidance*. We suppose this is more salient in reranking where the small batch size is common[3]. Then we validate their findings by training a CL model with fully half-precision loss computation, which yields degraded performance compared to precision-safe training (refer to Appendix B.2).

Back to our framework, $W$ controls the steps of model updates, or say the gradient scale. According to Chen et al. (2021), $W_{\text{CL}}$ should be small in the training process. And we expect $W_{\text{SFT}}$ to be larger than $W_{\text{CL}}$ to provide better optimization signal as SFT presents better performance. To verify this, we plot the $W$ of CL and SFT in training in Figure 3, where $W_{\text{CL}}$ indeed show relatively small values. SFT provides larger (better) $W$ than CL, thereby achieving stronger empirical performance. Equation 10 to 12 also shows that $W_{\text{SFT}}$ is larger than $W_{\text{CL}}$, since the denominator of $W_{\text{CL}}$ involves a sum of all negatives while the denominator of $W_{\text{SFT}}$ only adds up current instance.

Next, we investigate the fine-grained function of $W$. To create a cleaner analysis setting, we fix the direction in URL as $D_{\text{SFT}}$ unchanged, as it performs better. We first set weights of both positive and negatives to the fixed constant 1 as a baseline ($W_{base}$) following (Chen et al., 2021):

$$W^+ = 1, W_j^- = \frac{\exp(z(\mathbf{h}_j^-))}{\sum_j \exp(z(\mathbf{h}^-))}, \sum_j W_j^- = 1. \tag{14}$$

Although the earlier analysis suggests that the larger $W$ is preferable, this value 1 never appears in Figure 3, so we expect this setting to perform poorly. The experiment in Table 2 also align this.

---

[3]Consider a batch of instances, $\{O_1, \ldots, O_j\}$, is forward simultaneously during training with $k$ negatives per sample. While dense retrieval can achieve the negative size of $j \cdot (k+1)$ per instances, reranking models' are limited to $k + 1$. Furthermore, the increased number of input tokens at the reranking stage, compared to dense retrieval, imposes additional constraints on memory usage, resulting in a reduction in the value of negative size.

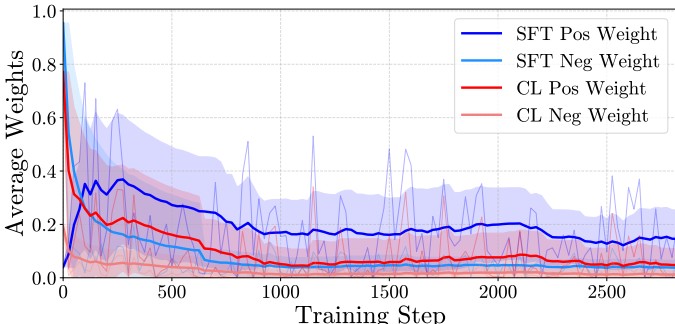

Figure 3: Evolution of positives and negatives average weights during training for SFT and CL.

| No. | Method | Avg. | Δ |
|-----|--------|------|---|
| 1 | $W_{\text{Base}}$ | 49.47 | - |
| 2 | + $\tau$ mask | 56.57 | ▲ 7.10 |
| 3 | + $W_{\text{CL}}$ | 56.23 | ▲ 6.76 |
| 4 | + $W_{\text{SFT}}$ | 58.19 | ▲ 8.72 |

Table 2: Evaluation of different properties of the weight component, where Δ denotes performance gain relative to $W_{\text{Base}}$.

Hence, we suppose that $W$ should be in a reasonable range. Meanwhile, the failure of constant $W$ indicates that instance-specific adjustment is necessary: *the model should update less on already-mastered instances and more on those it has not yet grasped.*

We adopt the predicted relevance scores $s$ as a guide and apply a masking rule: if a positive score is high enough, *i.e.,* $s(h_0) > 1 - \tau$, (or, conversely, a negative score is low enough, $s(h_j) < \tau$), we set $W^+ = 0$ (*resp.* $W_j^- = 0$) to halt further learning on that instance. In addition, we further set $W_{\text{CL}}$ and $W_{\text{SFT}}$ to the baseline and conduct training under the same conditions. The results are shown in Table 2, we can see that the simple masking rule can provide strong performance, comparable to CL. This indicates both CL and SFT follow the above instance-specific weight feature. More details of the experiment can be found in the Appendix B.2.

## 4.2 SEARCHING BETTER DIRECTION

Results in Table 1 indicate that the direction component also affects model performance, but it is not the dominant factor. Here we conduct additional experiments and try to find a better direction.

**Does adding more tokens improve performance?** SFT-based training is actually a binary classification on the token labels, where $D_{\text{SFT}}$ only involves "yes" and "no" tokens. One natural question is whether adding more tokens (*e.g.,* "true", "false", "maybe", *etc.* ) during training could improve the direction component and model performance? To investigate this, we randomly select 10,000 training instances and identify the top 16 tokens with the highest logits from the model's output, including "yes" and "no". For a comprehensive list of these tokens and details, please refer to the Appendix

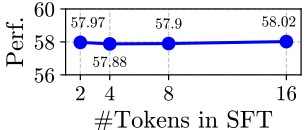

Figure 4: Results with different token numbers in SFT. The setting with 2 tokens is the standard SFT training.

B.3. We then train the model using this expanded token set. Figure 4 presents the results, which indicate that increasing the number of tokens does not significantly impact model performance. This result suggests that using only "yes" and "no" tokens is sufficient for effective SFT.

**Is it possible to learn a better direction?** The direction components, in essence, corresponds to the token embeddings of the LLM, which are pre-trained and keeping frozen during training. Before LLM, CL-based rerankers often learn a score-projection matrix from scratch. To see whether this still helps, we implement the random-initialized learnable weight $D_{\text{Rand.}}$ in URL. Table 3 shows that, for CL models, it does improve performance, yet still trails behind SFT. For SFT models, however, the strategy hurts performance. This is in line with the intuition: SFT is trained to predict the "yes/no" tokens, so replacing the weight with a randomly-initialized projection will loss the semantic signal from the pre-trained token embeddings.

| Weight | Direction | Perf. | Δ |
|--------|-----------|-------|---|
| $W_{\text{SFT}}$ | $D_{\text{SFT}}$ | 58.09 | - |
| | $D_{\text{Rand.}}$ | 56.75 | ▼ 1.34 |
| $W_{\text{CL}}$ | $D_{\text{CL}}$ | 56.40 | - |
| | $D_{\text{Rand.}}$ | 57.72 | ▲ 1.32 |

Table 3: Performance comparison of SFT and CL directions against random initialization $D_{\text{Rand.}}$.

| Model | Size | Single-Modal | | Cross-Modal | | | Fused-Modal | | | | Avg |
|---|---|---|---|---|---|---|---|---|---|---|---|
| | | T→T$_{(14)}$ | I→I$_{(1)}$ | T→I$_{(4)}$ | T→VD$_{(5)}$ | I→T$_{(5)}$ | T→IT$_{(2)}$ | IT→T$_{(4)}$ | IT→I$_{(2)}$ | IT→IT$_{(3)}$ | ALL$_{(40)}$ |
| GME-2B | 2.21B | 49.59 | 30.75 | 48.46 | 66.39 | 52.62 | 77.02 | 39.88 | 36.70 | 66.89 | 52.54 |
| *Qwen3* | 4.02B | 60.49 | – | – | – | – | – | – | – | – | – |
| *Jina-m0* | 2.21B | 55.36 | 27.50 | 59.46 | 73.13 | 55.43 | 74.95 | 27.82 | 37.65 | 51.54 | 54.36 |
| *MonoQwen* | 2.21B | 48.89 | 12.59 | 58.73 | 71.29 | 19.62 | 76.46 | 14.35 | 31.75 | 35.83 | 44.20 |
| GMR-3B | 3.75B | 59.22 | 29.76 | 58.85 | 72.38 | 63.06 | 81.96 | 48.81 | 43.97 | 79.08 | 61.40 |
| GMR-7B | 8.29B | 61.08 | 32.83 | 61.18 | 72.94 | 66.61 | 84.55 | 53.29 | 47.39 | 82.19 | 63.85 |

Table 4: Performance of different models on MRB. Each column corresponds to a task category, with the number of test sets indicated in parentheses. Evaluation metrics are provided in Appendix E.1. We adopt GME-2B as the retrieval backbone, while all other models rerank the top-100 retrieved candidates. ■ indicates the best result in reranking models, and ■ indicates the second-best.

## 5 EXPERIMENTS

### 5.1 SETTINGS

**Training Dataset** To develop a universal multimodal reranking model, we follow the settings of GME and curate training data from three categories: single-modal data (T→T, I→I), cross-modal data (I↔T, T→VD), and fused-modal data (IT↔T, IT→I, IT→IT). In total, we compile approximately **1.5** million training instances from diverse sources, including M-BEIR (Wei et al., 2025), ViDoRe (Faysse et al., 2025), ImageNet-1K (Deng et al., 2009), E-VQA (Mensink et al., 2023), and MS MARCO (Nguyen et al., 2016). To ensure fairness and efficiency in the comparative experiments reported in §4, we additionally construct a balanced and category-representative subset consisting of about 270K samples drawn from the full training dataset. The models, GMR-3B and GMR-7B, are trained on the complete dataset to achieve optimal performance, whereas the models evaluated in §4 are trained on the constructed subset. Details could be found in Appendix C.1.

**MRB Benchmark** To facilitate a more rigorous evaluation of model performance, we construct the MRB benchmark, which comprises **40** test datasets sourced from BEIR (Kamalloo et al., 2024), UMRB (Zhang et al., 2025b), ViDoRe (Faysse et al., 2025; Macé et al., 2025), and MIEB (Xiao et al., 2025). Collectively, these datasets span diverse modalities, domains, and task types, ensuring that the benchmark provides a comprehensive and representative assessment of model generalization. To more clearly highlight performance differences among models, we exclude test datasets on which GME-2B exhibits exceptionally high performance. A detailed description of the MRB benchmark composition is provided in Appendix C.2.

**Training Configuration** We adopt the Qwen2.5-VL-Instruct (Team, 2025) model series as the backbone of our multimodal large language model (MLLM), and conduct training at both 3-billion (3B) and 7-billion (7B) parameter scales. For efficient adaptation, we employ Low-Rank Adaptation (LoRA) with a rank of 16 and a learning rate of 1e-4. As evidenced by the comparative results in §4, within the domain of multimodal LLM reranking, SFT consistently outperforms CL. Consequently, we adopt SFT as the training strategy for our GMR series models.

During training, we set the maximum input length to 3,200 tokens. Each training sample is paired with 16 negative instances for the GMR-3B and GMR-7B models, and with 4 negative instances for the models mentioned in §4. Regarding the selection of negatives, we employ two strategies: *Random Selection* and *Hard Mining*, maintaining a balanced ratio of 1:1 between them. Further details on the negative sampling strategy are provided in Appendix C.3. To optimize GPU memory usage, we train the model using bfloat16 precision. All experiments were conducted on eight NVIDIA A100 GPUs, each equipped with 80 GB of memory.

**Baselines** We adopt GME-2B as the retrieval backbone to generate candidate results for each task. Specifically, the top-100 retrieved candidates are retained, and all reranking models are subsequently evaluated on this candidate pool. For the experiment described in §4, we reorder the top-25 candidates to balance fairness with efficiency. Our method is compared against three representative types

of reranking systems: (1) A representative textual model : Qwen3-Reranker (Zhang et al., 2025d) (*Qwen3*), exemplifying recent advancements in text-based reranking. (2) A versatile multimodal reranking model: Jina-rerank-m0[4](*Jina-m0*). This model natively supports single-modal tasks and cross-modal tasks. Leveraging the flexibility of its MLLM architecture, we extend its application to fused-modal tasks by adopting its input template. The specifics of these adaptations are detailed in Appendix D.4. (3) A cutting-edge visual document reranking model: MonoQwen2-VL-v0.1 (Chaffin & Lac, 2024) (*MonoQwen*). Similar to our approach with Jina-rerank-m0, we evaluate this model across all task types. The input templates used is provided in Appendix D.5.

This comprehensive evaluation benchmarks our method against leading models across diverse modalities and task types, enabling a thorough assessment of its effectiveness.

## 5.2 MAIN RESULTS

We first examine the effect of the number of negatives. In SFT, where query–candidate similarity is formulated as a binary classification task, the number of negatives directly affects model performance. To identify an appropriate setting under our computational budget, we experiment with varying numbers of negatives (Figure 5). Performance consistently improves with more negatives, peaking at 16. Moreover, SFT outperforms CL across all settings (Appendix E.2). Based on these results, we set the number of negatives to 16 in training. Given the impact of random initialization on performance (§4.2), we also conduct an ablation on freezing the LM head (Appendix E.3) and find that has no effect on SFT performance.

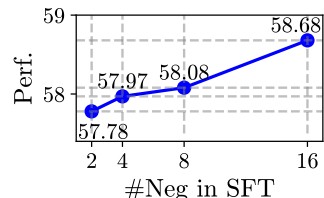

Figure 5: Results with different numbers of negative in SFT.

We next examine the evaluation results. Table 4 presents a comprehensive overview of the baseline systems' performance. The reported scores are averaged across the respective sub-tasks and are organized according to the retrieval modality: Single-Modal, Cross-Modal, and Fused-Modal. For completeness, the overall micro-average score across all sub-tasks is provided in the final column.

**Achieve state-of-the-art performance in universal multimodal reranking.** Analyzing the average metrics, our smaller model, GMR-3B, exhibits superior results compared to the fused-modal reranking model (Jina-Rerank-m0). The larger GMR-7B further elevates this performance, underscoring the efficacy in addressing universal multimodal reranking challenges.

**Rival and surpass leading textual reranker.** We conduct a comparative analysis with the state-of-the-art textual reranking model, Qwen3-Reranker, which is specifically optimized for the T→T task within the Single-Modal category and comprises approximately 4 billion parameters. Our smaller model exhibited similar performance metrics when evaluated against models of similar parameter scale. Notably, our larger model surpass the performance of Qwen3-Reranker, providing strong empirical evidence for the efficacy of our proposed methodology.

**Maintain strong robustness as reranking depth increases.** Recent studies (Jacob et al., 2025) suggest that reranker performance may degrade as the number of documents increases. To examine this effect, we evaluate GMR and key baselines under the top-25 and top-100 settings, with results shown in the Figure 6. In T→T tasks, all models improve when moving from top-25 to top-100, and the gains correlate with model strength—stronger models benefit more. On the full MRB benchmark, our GMR models also exhibit consistent improvements (1.47% / 2.39%), whereas Jina-Rerank-m0 shows a slight drop (-0.08%). These results indicate that our models maintain stronger robustness across different top-N configurations.

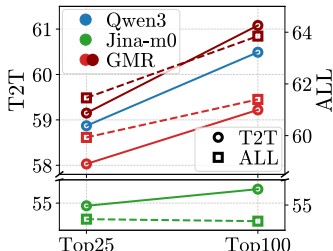

Figure 6: Performance comparison under top-25 and top-100 reranking settings.

**Adapt seamlessly to visual-document reranking.** In Table 4, we compare with the visual document reranking model, MonoQwen2-VL-v0.1, which is specifically tailored for the T→VD task. Our proposed models demonstrate performance metrics that are surpass those of this task-specific baseline, which suggests a promising direction for developing

---

[4]https://huggingface.co/jinaai/jina-reranker-m0

| Task | Model | Art | Med. | Sci. | Hum. | Avg. |
|------|-------|-----|------|------|------|------|
| Retrieval | GME-2B | 73.86 | 46.99 | 62.69 | 56.16 | 59.93 |
| Rerank | *MonoQwen* | 57.49 | 42.81 | 47.55 | 44.36 | 48.05 |
|  | *jina-m0* | 77.91 | 55.21 | 70.05 | 66.35 | 67.38 |
| Top-100 | GMR-3B | **82.38** | **59.47** | **72.14** | **68.79** | **70.69** |
|  | GMR-7B | **82.91** | **63.93** | **75.90** | **74.14** | **74.22** |

Table 5: MRMR Knowledge Performance across Models (reported using NDCG@10). ■ indicates the best result in reranking models, and ■ indicates the second-best.

more efficient and adaptable information re-reanking systems that can seamlessly handle diverse modalities within a single architecture.

## 5.3 ADDITIONAL EXPERIMENTS IN MULTI-IMAGE SETTING

To evaluate model performance beyond the setting where queries and documents each contain a single image, we further assess our models on the MRMR(Zhang et al., 2026). The results are presented in the Table 5. In the Knowledge subset of MRMR, queries are multimodal—containing both text and images—while documents cover the full modality space, including pure-text documents as well as mixed text–image documents with either single or multiple images. This design enables a comprehensive examination of model robustness and multimodal reasoning capabilities.

As shown in the Table 5, the GMR series achieves the strongest performance among all reranking models and delivers substantial improvements over the retrieval model (GME-2B). These results demonstrate that our approach remains effective in multi-image scenarios and is capable of achieving superior performance under more complex multimodal conditions.

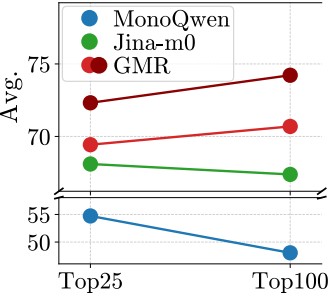

In addition, we evaluate the average performance of the models under the top-25 and top-100 settings, as shown in the Figure 7. Both Jina-rerank-m0 and MonoQwen2-VL-v0.1 exhibit performance drops when moving from top-25 to top-100, whereas our models (GMR-3B / GMR-7B) achieve consistent gains (+1.26% / +1.90%). These results demonstrate that our approach remains effective in fully multimodal scenarios, including cases with multiple images, and continues to deliver superior performance under more challenging top-N configurations.

Figure 7: Average Performance comparison under top-25 and top-100 reranking settings in the Knowledge subset of MRMR.

## 6 CONCLUSION

In this work, we show that supervised fine-tuning (SFT) consistently outperforms contrastive learning (CL) for LLM-based reranking. By decomposing the loss into *weight* and *direction* components, we find that the weight term primarily drives performance gains by strengthening optimization signals and providing input-specific guidance. While SFT's directional component is nearly optimal, CL requires learning a score-projection matrix to achieve comparable results. Building on these insights, we develop the GMR-3B and GMR-7B models, which set new state-of-the-art results on the MRB benchmark covering 40 datasets. By releasing MRB, our models, and code, we provide a solid foundation for future research in large-scale multimodal retrieval and universal LLM reranking, underscoring both methodological and practical significance.

## ACKNOWLEDGMENTS

We thank the anonymous reviewers and chairs for their valuable feedback and suggestions. This work receives support from the Natural Science Foundation of China (under Grants 62336008, 624B2048), Research Grant Council of Hong Kong (PolyU/15209724), and the Shenzhen Basic Research Program (Grant No. JCYJ20240813105111016).

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

APPENDIX

# A  METHOD DETAILS

## A.1  GMR INPUT TEMPLATE

Following a chat-based template, The prompt formulates a binary classification task by providing the model with a specific Instruction, Query, and Document for evaluation as shon in Figure 8.

> <|im_start|>system:
> **Judge whether the Document meets the requirements based on the Query and the Instruct provided. Note that the answer can only be "yes" or "no". <|im_end|>**
> **<|im_start|> user :**
> **<Instruction>: {Instruction}**
> **<Query>: {Query}**
> **<Document>: {Document} <|im_end|>**
> **<|im_start|> assistant :**

Figure 8: The structured input template for GMR series models.

## A.2  LOSS FUNCTION DECOMPOSITION

In this section, we elaborate on the derivation process of the equation in §3.3.

- Equation 6:

$$
\begin{aligned}
-\frac{\partial \mathcal{L}^{\mathrm{CL}}}{\partial \mathbf{h}_0^+} &= -\frac{\partial \mathcal{L}^{\mathrm{CL}}}{\partial z^y(\mathbf{h}_0^+)} \frac{\partial z^y(\mathbf{h}_0^+)}{\partial \mathbf{h}_0^+} \\
&= -\frac{\partial(-\log \frac{\exp(s(ins,q,d_0^+))}{\exp(s(ins,q,d_0^+))+\sum_i \exp(s(ins,q,d_i^-))})}{\partial z^y(\mathbf{h}_0^+)} \frac{\partial z^y(\mathbf{h}_0^+)}{\partial \mathbf{h}_0^+} \\
&= \frac{\partial(\log \frac{\exp(z^y(\mathbf{h}_0^+))}{\exp(z^y(\mathbf{h}_0^+))+\sum_i \exp(z^y(\mathbf{h}_i^-))})}{\partial z^y(\mathbf{h}_0^+)} \frac{\partial z^y(\mathbf{h}_0^+)}{\partial \mathbf{h}_0^+} \\
&= \frac{\exp(z^y(\mathbf{h}_0^+))+\sum_i \exp(z^y(\mathbf{h}_i^-))}{\exp(z^y(\mathbf{h}_0^+))} \cdot \frac{\partial(\frac{\exp(z^y(\mathbf{h}_0^+))}{\exp(z^y(\mathbf{h}_0^+))+\sum_i \exp(z^y(\mathbf{h}_i^-))})}{\partial z^y(\mathbf{h}_0^+)} \cdot \frac{\partial z^y(\mathbf{h}_0^+)}{\partial \mathbf{h}_0^+} \\
&= \frac{\exp(z^y(\mathbf{h}_0^+))+\sum_i \exp(z^y(\mathbf{h}_i^-))}{\exp(z^y(\mathbf{h}_0^+))} \cdot \frac{\sum_i \exp(z^y(\mathbf{h}_i^-))}{(\exp(z^y(\mathbf{h}_0^+))+\sum_i \exp(z^y(\mathbf{h}_i^-)))^2} \\
&\quad \cdot \frac{\partial \exp(z^y(\mathbf{h}_0^+))}{\partial z^y(\mathbf{h}_0^+)} \cdot \frac{\partial z^y(\mathbf{h}_0^+)}{\partial \mathbf{h}_0^+} \\
&= \frac{\exp(z^y(\mathbf{h}_0^+))+\sum_i \exp(z^y(\mathbf{h}_i^-))}{\exp(z^y(\mathbf{h}_0^+))} \cdot \frac{\sum_i \exp(z^y(\mathbf{h}_i^-))}{(\exp(z^y(\mathbf{h}_0^+))+\sum_i \exp(z^y(\mathbf{h}_i^-)))^2} \\
&\quad \cdot \exp(z^y(\mathbf{h}_0^+)) \cdot \frac{\partial z^y(\mathbf{h}_0^+)}{\partial \mathbf{h}_0^+} \\
&= \frac{\sum_j \exp(z^y(\mathbf{h}_j^-))}{\exp(z^y(\mathbf{h}_0^+))+\sum_j \exp(z^y(\mathbf{h}_j^-))} \frac{\partial z^y(\mathbf{h}_0^+)}{\partial \mathbf{h}_0^+} \\
&= \frac{\sum_j \exp(z^y(\mathbf{h}_j^-))}{\exp(z^y(\mathbf{h}_0^+))+\sum_j \exp(z^y(\mathbf{h}_j^-))} \mathcal{M}_y
\end{aligned} \tag{15}
$$

- Equation 7:

$$
\begin{aligned}
-\frac{\partial \mathcal{L}^{\mathrm{CL}}}{\partial \mathbf{h}_i^-} &= -\frac{\partial \mathcal{L}^{\mathrm{CL}}}{\partial z^y(\mathbf{h}_i^-)}\frac{\partial z^y(\mathbf{h}_i^-)}{\partial \mathbf{h}_i^-}\\
&= -\frac{\partial\left(-\log \frac{\exp(s(ins,q,d_0^+))}{\exp(s(ins,q,d_0^+))+\sum_i \exp(s(ins,q,d_i^-))}\right)}{\partial z^y(\mathbf{h}_i^-)}\frac{\partial z^y(\mathbf{h}_i^-)}{\partial \mathbf{h}_i^-}\\
&= \frac{\partial\left(\log \frac{\exp(z^y(\mathbf{h}_0^+))}{\exp(z^y(\mathbf{h}_0^+))+\sum_i \exp(z^y(\mathbf{h}_i^-))}\right)}{\partial z^y(\mathbf{h}_i^-)}\frac{\partial z^y(\mathbf{h}_i^-)}{\partial \mathbf{h}_i^-}\\
&= \frac{\exp(z^y(\mathbf{h}_0^+))+\sum_i \exp(z^y(\mathbf{h}_i^-))}{\exp(z^y(\mathbf{h}_0^+))}\cdot\left(-\frac{\exp(z^y(h_0^+))}{(\exp(z^y(\mathbf{h}_0^+))+\sum_i \exp(z^y(\mathbf{h}_i^-)))^2}\right)\\
&\quad \cdot \exp(z^y(\mathbf{h}_i^-))\cdot\frac{\partial z^y(\mathbf{h}_i^-)}{\partial \mathbf{h}_0^+}\\
&= -\frac{\exp(z^y(\mathbf{h}_i^-))}{\exp(z^y(\mathbf{h}_0^+))+\sum_j \exp(z^y(\mathbf{h}_i^-))}\frac{\partial z^y(\mathbf{h}_i^-)}{\partial \mathbf{h}_i^-}\\
&= -\frac{\exp(z^y(\mathbf{h}_i^-))}{\exp(z^y(\mathbf{h}_0^+))+\sum_j \exp(z^y(\mathbf{h}_i^-))}\mathcal{M}_y
\end{aligned}
\tag{16}
$$

- Equation 8:

$$
\begin{aligned}
-\frac{\partial \mathcal{L}^{\mathrm{SFT}}}{\partial \mathbf{h}_0^+} &= -\frac{\partial \mathcal{L}^{\mathrm{SFT}}}{\partial z^y(\mathbf{h}_0^+)}\frac{\partial z^y(\mathbf{h}_0^+)}{\partial \mathbf{h}_0^+}-\frac{\partial \mathcal{L}^{\mathrm{SFT}}}{\partial z^n(\mathbf{h}_0^+)}\frac{\partial z^n(\mathbf{h}_0^+)}{\partial \mathbf{h}_0^+}\\
&= -\frac{\partial\left(-\log(p(\text{``yes''}|z(\{\text{``yes''},\text{``no''}\}|\{ins,q,d_i\}))))\right)}{\partial z^y(\mathbf{h}_0^+)}\frac{\partial z^y(\mathbf{h}_0^+)}{\partial \mathbf{h}_0^+}\\
&\quad -\frac{\partial\left(-\log(p(\text{``yes''}|z(\{\text{``yes''},\text{``no''}\}|\{ins,q,d_i\}))))\right)}{\partial z^n(\mathbf{h}_0^+)}\frac{\partial z^n(\mathbf{h}_0^+)}{\partial \mathbf{h}_0^+}\\
&= \frac{\partial\left(\log \frac{e^{z(\text{``yes''}|\{ins,q,d\})}}{e^{z(\text{``yes''}|\{ins,q,d\})}+e^{z(\text{``no''}|\{ins,q,d\})}}\right)}{\partial z^y(\mathbf{h}_0^+)}\frac{\partial z^y(\mathbf{h}_0^+)}{\partial \mathbf{h}_0^+}+\\
&\quad \frac{\partial\left(\log \frac{e^{z(\text{``yes''}|\{ins,q,d\})}}{e^{z(\text{``yes''}|\{ins,q,d\})}+e^{z(\text{``no''}|\{ins,q,d\})}}\right)}{\partial z^n(\mathbf{h}_0^+)}\frac{\partial z^n(\mathbf{h}_0^+)}{\partial \mathbf{h}_0^+}\\
&= \frac{\partial\left(\log \frac{\exp(z^y(\mathbf{h}_0^+))}{\exp(z^y(\mathbf{h}_0^+))+\exp(z^n(\mathbf{h}_0^+))}\right)}{\partial z^y(\mathbf{h}_0^+)}\frac{\partial z^y(\mathbf{h}_0^+)}{\partial \mathbf{h}_0^+}\\
&\quad +\frac{\partial\left(\log \frac{\exp(z^y(\mathbf{h}_0^+))}{\exp(z^y(\mathbf{h}_0^+))+\exp(z^n(\mathbf{h}_0^+))}\right)}{\partial z^n(\mathbf{h}_0^+)}\frac{\partial z^n(\mathbf{h}_0^+)}{\partial \mathbf{h}_0^+}\\
&= \frac{\exp(z^n(\mathbf{h}_0^+))}{\exp(z^y(\mathbf{h}_0^+))+\exp(z^n(\mathbf{h}_0^+))}\frac{\partial z^y(\mathbf{h}_0^+)}{\partial \mathbf{h}_0^+}\\
&\quad -\frac{\exp(z^n(\mathbf{h}_0^+))}{\exp(z^y(\mathbf{h}_0^+))+\exp(z^n(\mathbf{h}_0^+))}\frac{\partial z^n(\mathbf{h}_0^+)}{\partial \mathbf{h}_0^+}\\
&= \frac{\exp(z^n(\mathbf{h}_0^+))}{\exp(z^y(\mathbf{h}_0^+))+\exp(z^n(\mathbf{h}_0^+))}\left(\frac{\partial z^y(\mathbf{h}_0^+)}{\partial \mathbf{h}_0^+}-\frac{\partial z^n(\mathbf{h}_0^+)}{\partial \mathbf{h}_0^+}\right)\\
&= \frac{\exp(z^n(\mathbf{h}_0^+))}{\exp(z^y(\mathbf{h}_0^+))+\exp(z^n(\mathbf{h}_0^+))}(\mathcal{M}_{\mathrm{y}}-\mathcal{M}_{\mathrm{n}})
\end{aligned}
\tag{17}
$$

- Equation 9:

$$
\begin{aligned}
-\frac{\partial \mathcal{L}^{\text{SFT}}}{\partial \mathbf{h}_i^-} &= -\frac{\partial \mathcal{L}^{\text{SFT}}}{\partial z^y(\mathbf{h}_j^-)}\frac{\partial z^y(\mathbf{h}_i^-)}{\partial \mathbf{h}_i^-} - \frac{\partial \mathcal{L}^{\text{SFT}}}{\partial z^n(\mathbf{h}_i^-)}\frac{\partial z^n(\mathbf{h}_i^-)}{\partial \mathbf{h}_i^-} \\
&= -\frac{\partial(-\log(p(\text{``no''}|z(\{\text{``yes''},\text{``no''}\}|\{ins,q,d_i\}))))}{\partial z^y(\mathbf{h}_i^-)}\frac{\partial z^y(\mathbf{h}_i^-)}{\partial \mathbf{h}_i^-} \\
&\quad -\frac{\partial(-\log(p(\text{``no''}|z(\{\text{``yes''},\text{``no''}\}|\{ins,q,d_i\}))))}{\partial z^n(\mathbf{h}_i^-)}\frac{\partial z^n(\mathbf{h}_i^-)}{\partial \mathbf{h}_i^-} \\
&= \frac{\partial(\log \frac{e^{z(\text{``no''}|\{ins,q,d\})}}{e^{z(\text{``yes''}|\{ins,q,d\})}+e^{z(\text{``no''}|\{ins,q,d\})}})}{\partial z^y(\mathbf{h}_i^-)}\frac{\partial z^y(\mathbf{h}_i^-)}{\partial \mathbf{h}_i^-} + \\
&\quad \frac{\partial(\log \frac{e^{z(\text{``no''}|\{ins,q,d\})}}{e^{z(\text{``yes''}|\{ins,q,d\})}+e^{z(\text{``no''}|\{ins,q,d\})}})}{\partial z^n(\mathbf{h}_i^-)}\frac{\partial z^n(\mathbf{h}_i^-)}{\partial \mathbf{h}_i^-} \\
&= \frac{\partial(\log \frac{\exp(z^n(\mathbf{h}_i^-))}{\exp(z^y(\mathbf{h}_i^-))+\exp(z^n(\mathbf{h}_i^-))})}{\partial z^y(\mathbf{h}_i^-)}\frac{\partial z^y(\mathbf{h}_i^-)}{\partial \mathbf{h}_i^-} \\
&\quad + \frac{\partial(\log \frac{\exp(z^n(\mathbf{h}_i^-))}{\exp(z^y(\mathbf{h}_i^-))+\exp(z^n(\mathbf{h}_i^-))})}{\partial z^n(\mathbf{h}_i^-)}\frac{\partial z^n(\mathbf{h}_i^-)}{\partial \mathbf{h}_i^-} \\
&= -\frac{\exp(z^y(\mathbf{h}_i^-))}{\exp(z^y(\mathbf{h}_i^-)) + \exp(z^n(\mathbf{h}_i^-))}\left(\frac{\partial z^y(\mathbf{h}_i^-)}{\partial \mathbf{h}_i^-} - \frac{\partial z^n(\mathbf{h}_i^-)}{\partial \mathbf{h}_i^-}\right) \\
&= -\frac{\exp(z^y(\mathbf{h}_i^-))}{\exp(z^y(\mathbf{h}_i^-)) + \exp(z^n(\mathbf{h}_i^-))}(\mathcal{M}_y - \mathcal{M}_n) \quad\quad (18)
\end{aligned}
$$

# B  ANALYSIS EXPERIMENT

## B.1  THE INFLUENCE OF PRECISION ON CL

We validate the findings of FlatNCE by performing full half-precision training during loss function computation on the contrastive learning (CL) model. Specifically, we configure the model to use BF16 for accuracy, and in the loss computation process (refer to Algorithm 1), we control all other variables while varying the precision of the weight computations between FP16 and FP32 to assess their impact on model performance. The results show that FP32 precision yields better performance than FP16 precision, confirming that computational precision significantly affects the effectiveness of contrastive learning.

| Method | Precision | Avg | $\Delta$ |
|--------|-----------|-------|------|
| CL | FP16 | 56.09 | - |
|    | FP32 | 56.40 | ▲ 0.31 |

Table 6: Impact of precision on Contrastive Learning's performance.

## B.2  FUNCTION OF WEIGHT

To investigate the role of the weight, we first define $s(\mathbf{h}_i) = \frac{\exp(z^y(h_i))}{\exp(z^y(h_i))+\exp(z^n(h_i))}$. Since $s(h_i)$ is bounded within $[0, 1]$, prior experience with embedding models suggests that an appropriate scaling factor is necessary to accelerate model convergence. Therefore, we introduce a temperature parameter $\beta = 5 \times 10^{-2}$ into Equation 14, yielding $W_j^- = \frac{\exp(s(\mathbf{h}_j^-)/\beta)}{\sum_j \exp(s(\mathbf{h}^-)/\beta)}$. In addition, for experiments involving the masking rule, we vary $\tau \in 10^{-2}, 10^{-3}, 10^{-4}$ to identify the configuration that achieves optimal performance. For the experiment with $W_{CL}$, we follow Equation 10 and 11, consistent with the requirements of contrastive learning, where the positive and negative weights must satisfy the constraint $W^+ = \sum W^-$. Since directly setting $W_{+W_{CL}} = W_{Base}W_{CL}$ would violate this condition, we instead use $W_{+W_{CL}} =$

| Method | $\tau$ | Avg |
|--------|------|-------|
| w/ $\tau$ mask | 1e-2 | 55.07 |
|                | 1e-3 | 56.57 |
|                | 1e-4 | 55.89 |

Table 7: The performance of the model under different values of $\tau$.

$W_{CL}$ for comparison with $W_{Base}$. For the experiment with $W_{SFT}$, we aim to demonstrate that $W_{SFT}$ can effectively enhance the performance of $W_{Base}$. Following Equation 12 and 13, we set $W_{+W_{SFT}} = W_{Base}W_{SFT}$ and evaluate its impact on model performance.

### B.3 THE INFLUENCE OF TOKEN SELECTION

To examine whether introducing additional tokens during training can enhance the directionality component and improve model performance, we randomly sample 10,000 instances together with their corresponding positives and negatives. Based on the model outputs, we identify the top 16 tokens with the highest average logits, which include "yes" and "no." The remaining tokens in this set are: {"No," "Yes," "NO," "YES," "The," "None," "In," "Answer," "This," "To," "Not," "not," "There," "-no"}.

## C EXPERIMENT SETTING

### C.1 TRAINING DATASETS

Our training dataset is curated from diverse sources, including M-BEIR, ViDoRe, ImageNet-1K, E-VQA, and Ms Marco. These datasets cover a wide array of domains, ensuring that the model is exposed to varied and representative examples across different tasks. To ensure balanced representation across task domains, we sample 100k instances from ImageNet-1K and integrated them into our training corpus.

In total, our training dataset consists of approximately 1.5 million instances, which are distributed across various domains to ensure robust learning. The detailed distribution of the data across these domains is carefully visualized in Figure 9.

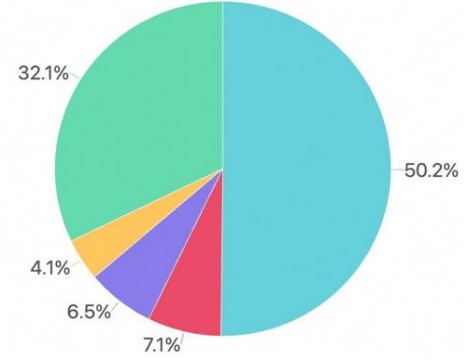

Figure 9: The proportion of the training data.

To ensure a fair comparison between supervised fine-tuning and contrastive learning, we construct a balanced, category-representative subset of approximately 270K samples from our training dataset, and the details could be found in Table 8.

| Class | Task | Datasets | Number |
|---|---|---|---|
| Single-Modal(4) | T→T (2) | WebQA† Ms Marco | 30000 |
| | I→I (2) | Nights† ImageNet-1K | 30000 |
| Cross-Modal(6) | T→I (2) | Fashion200k† VisualNews† | 29958 |
| | T→VD (1) | ViDoRe | 30000 |
| | I→T (3) | Fashion200k† MScoco† VisualNews† | 30882 |
| Fused-Modal(11) | T→IT (2) | EDIS† WebQA† | 30000 |
| | IT→T (3) | LLava† OVEN† Remuq† | 30382 |
| | IT→I (2) | CIRR† FashionIQ† | 29528 |
| | IT→IT (3) | E-VQA OVEN† | 30000 |

Table 8: The details of sub trainset. † means that they belong to the M-BEIR dataset.

## C.2 MRB BENCHMARK

Since overly simple tasks fail to effectively differentiate the performance of various rerank models, we exclude the dataset on which the GME-2B model achieves exceptionally high performance. Detailed descriptions of MRB Benchmark are provided in Tables 9 and 10.

| Class | Task | Datasets |
|---|---|---|
| Single-Modal(15) | T→T (14) | ArguAna† Climate-FEVER† CQADupStack† DBPedia† FIQA2018† HotpotQA† MSMARCO† NFCorpus† NQ† Quora† SCIDOCS† SciFact† Touche2020† TRECCOVID† |
| | I→I (1) | Nights* |
| Cross-Modal(14) | T→I (4) | VisualNews* Fashion200k* Memotion* HatefulMemes* |
| | T→VD (5) | TAT-DQA$^\dagger$ ArxivQA$^\dagger$ DocVQA$^\dagger$ MIT Tissue Interaction$^\dagger$ World Economic Reports$^\dagger$ |
| | I→T (5) | VisualNews* Fashion200K* Memotion* GLDv2* HatefulMemes* |
| Fused-Modal(11) | T→IT (2) | WebQA* EDIS* |
| | IT→T (4) | OVEN* INFOSEEK* OKVQA* VizWiz* |
| | IT→I (2) | FashionIQ* CIRR* |
| | IT→IT (3) | OVEN* E-VQA* INFOSEEK* |

Table 9: An overview of datasets in *MRB*. † means it belong to BEIR. * means it belong to UMRB. $^\dagger$ means it belong to ViDoRe. * means it belong to MIEB.

## C.3 NEGATIVE SELECTION

The quality and diversity of negatives greatly affect the final performance of the reranker. Overly simple negatives can make the model lack the ability to distinguish hard negatives from positives, while overly difficult documents are very likely to be false negatives that give the model incorrect update signal. Therefore, we adopt two strategies to select negatives: **(1) Random Selection**. Randomly select irrelevant document as negatives to enhance the generalization ability of the model. **(2) Hard Mining**. For each query in every dataset, we use GME-2B to search for the corresponding documents to obtain the top 100, and randomly select $k$ irrelevant samples from them as hard negatives to improve the reranking performance. We employ this set of hard negatives for all the models trained in this paper. While training, we always maintain the ratio of random negatives to hard negatives at 1:1 to balance the diversity and quality of the data.

# D MODEL SETTINGS

## D.1 GME-2B

We employ the GME-2B model as the foundational retrieval model, generating the initial retrieval results that serve as the input to our diverse reranking approaches. Recognizing that the GME series models leverage instruction fine-tuning, we incorporate task-specific instructions into the input query to enhance the retrieval model's performance.

Aligning with the UMRB benchmark, we curate the specific instructions for each task, as comprehensively detailed in Table 14.

## D.2 QWEN3-RERANKER

Paralleling our approach, Qwen3-Reranker leverages Large Language Models for point-wise reranking within a singular contextual framework. To facilitate instruction-following capabilities, the model incorporates task-specific instructions directly into the input context. By utilizing the LLM's inherent chat template, the similarity assessment is reframed as a binary classification paradigm.

| Name | Type | Categ. | Eval Samples | Candidates Nums | Eval Query avg. chars | Eval Candidate avg. chars |
|---|---|---|---|---|---|---|
| ArguAna | Single-Modal | T→T | 1406 | 8,674 | 192.98 | 166.80 |
| Climate-FEVER | Single-Modal | T→T | 1,535 | 5,416,593 | 20.13 | 84.76 |
| CQADupStack | Single-Modal | T→T | 13,145 | 457,199 | 8.59 | 129.09 |
| DBPedia | Single-Modal | T→T | 400 | 4,635,922 | 5.39 | 49.68 |
| FiQA2018 | Single-Modal | T→T | 648 | 57,638 | 10.77 | 132.32 |
| HotpotQA | Single-Modal | T→T | 7,405 | 5,233,329 | 17.61 | 46.30 |
| MSMARCO | Single-Modal | T→T | 6,980 | 8,841,823 | 5.96 | 55.98 |
| NFCorpus | Single-Modal | T→T | 323 | 3,633 | 3.30 | 232.26 |
| NQ | Single-Modal | T→T | 3,452 | 2,681,468 | 9.16 | 78.88 |
| Quora | Single-Modal | T→T | 10,000 | 522,931 | 9.53 | 11.44 |
| SCIDOCS | Single-Modal | T→T | 1,000 | 25,657 | 9.38 | 176.19 |
| SciFact | Single-Modal | T→T | 300 | 5,183 | 12.37 | 213.63 |
| Touche2020 | Single-Modal | T→T | 49 | 382,545 | 6.55 | 292.37 |
| TRECCOVID | Single-Modal | T→T | 50 | 171,332 | 10.60 | 160.77 |
| Nights | Single-Modal | I→I | 2,120 | 40,038 | - | - |
| VisualNews | Cross-Modal | T→I | 19,995 | 542,246 | 18.78 | - |
| Fashion200k | Cross-Modal | T→I | 1,719 | 201,824 | 4.89 | - |
| HatefulMemes | Cross-Modal | T→I | 1000 | 10000 | 10.42 | - |
| Memotion | Cross-Modal | T→I | 697 | 6988 | 14.77 | - |
| TAT-DQA | Cross-Modal | T→VD | 1,646 | 277 | 12.44 | - |
| ArxivQA | Cross-Modal | T→VD | 500 | 500 | 17.12 | - |
| DocVQA | Cross-Modal | T→VD | 451 | 500 | 8.23 | - |
| WER | Cross-Modal | T→VD | 58 | 452 | 13.05 | - |
| MITTI | Cross-Modal | T→VD | 160 | 1016 | 13.91 | - |
| VisualNews | Cross-Modal | I→T | 20,000 | 537,568 | - | 18.53 |
| Fashion200k | Cross-Modal | I→T | 4,889 | 61,707 | - | 4.95 |
| GLDv2 | Cross-Modal | I→T | 1704 | 674 | - | 3.18 |
| Memotion | Cross-Modal | T→I | 697 | 6988 | - | 14.67 |
| HatefulMemes | Cross-Modal | I→T | 1000 | 10000 | - | 11.53 |
| WebQA | Fused-Modal | T→IT | 2,511 | 403,196 | 16.43 | 12.83 |
| EDIS | Fused-Modal | T→IT | 3,241 | 1,047,067 | 20.07 | 15.53 |
| OVEN | Fused-Modal | IT→T | 50,004 | 676,667 | 6.52 | 82.13 |
| INFOSEEK | Fused-Modal | IT→T | 11,323 | 611,651 | 8.76 | 91.49 |
| OKVQA | Fused-Modal | IT→T | 5,046 | 114,516 | 8.09 | 102.55 |
| VizWiz | Fused-Modal | IT→T | 4319 | 2091 | 7.17 | - |
| FashionIQ | Fused-Modal | IT→I | 6,003 | 74,381 | 11.70 | - |
| CIRR | Fused-Modal | IT→I | 4,170 | 21,551 | 11.01 | - |
| OVEN | Fused-Modal | IT→IT | 14,741 | 335,135 | 5.91 | 94.76 |
| EVQA | Fused-Modal | IT→IT | 3,743 | 68,313 | 9.38 | 211.12 |
| INFOSEEK | Fused-Modal | IT→IT | 17,593 | 481,782 | 7.94 | 96.00 |

Table 10: Tasks in *MRB*. Following UMRB, We count the number of datasets under each task type, the number of evaluation instances, the size of the candidate set, and the average length of the text.

Specifically, for $T \to T$ tasks, we set task-specific instructions the same as GME, as comprehensively illustrated in Table 14.

## D.3 GMR

In our GMR series models, we incorporate the retrieval instructions into the input context, yielding two advantages. Primarily, this approach eliminates the need for task-specific instruction redesign at the reranking stage, enabling seamless instruction transfer from the retrieval phase.

Moreover, by strategically integrating instructions into the contextual input, we effectively guide the model's comprehension, facilitating enhanced task understanding and robust instruction-following capabilities. The comprehensive instruction sets for both training and testing phases are meticulously detailed in Tables 11 and 14, respectively.

## D.4 JINA-RERANK-M0

Jina-rerank-m0 demonstrates inherent capabilities for processing single-modal and cross-modal tasks. By leveraging the architectural flexibility of Multimodal Large Language Model framework, we extend its operational scope to encompass fused-modal tasks through a input template adaptation.

| Task | Dataset | Query Instruction |
|------|---------|-------------------|
| T→T | WebQA
Ms Marco | Retrieve passages from Wikipedia that provide answers to the following question.
Given a question, retrieve relevant passages that answer the question. |
| I→I | Nights
ImageNet-1K | Find a day-to-day image that looks similar to the provided image.
Retrieve images of the same type as the one in the question. |
| T→I | Fashion200k
VisualNews | Based on the following fashion description, retrieve the best matching image.
Identify the news-related image in line with the described event. |
| T→VD | ViDoRe | Find a screenshot that relevant to the user's question. |
| I→T | VisualNews
Fashion200k
MSCOCO | Find a caption for the news in the given photo.
Find a product description for the fashion item in the image.
Find an image caption describing the following everyday image. |
| T→IT | WebQA
EDIS | Find a Wikipedia image that answers this question.
Find a news image that matches the provided caption. |
| IT→T | OVEN
LLava
Remuq | Retrieve a Wikipedia paragraph that provides an answer to the given query about the image.
Provide a specific decription of the image along with the following question.
Retrieve a fact-based paragraph that provides an answer to the given query about the image. |
| IT→I | FashionIQ
CIRR | Find a fashion image that aligns with the reference image and style note.
Retrieve a day-to-day image that aligns with the modification instructions of the provided image. |
| IT→IT | OVEN
E-VQA | Retrieve a Wikipedia image-description pair that provides evidence for the question of this image.
Determine the Wikipedia image-snippet pair that matches my question about this image. |

Table 11: The instructions for training dataset. We set the instructions for the GMR series models on each task during training as shown in the Table.

For text and image-modal inputs, Jina-rerank-m0 organizes Query/Document configurations, as comprehensively illustrated in Table 12. Building upon this foundational template, we design a input organization strategy for fused-modal scenarios, represented in the **Fused** configuration.

Ultimately, the model's input is standardized to the canonical format:"{**Document**}\n{**Query**}".

| | Query | Document |
|------|-------|----------|
| **Text** | `**Query**:\n{query}` | `**Document**:\n {doc}` |
| **Image** | `**Query**:`
`<vision_start><image_pad><vision_end>` | `**Document**:`
`<vision_start><image_pad><vision_end>` |
| **Fused** | `**Query**:`
`<vision_start><image_pad><vision_end>{query}` | `**Document**:`
`<vision_start><image_pad><vision_end>{doc}` |

Table 12: The input template of Jina-rerank-m0. We refer to it's format settings for **Text** and **Image** to set the input format of fused-modal data, then format the input as "{**Document**}\n{**Query**}".

## D.5 MONOQWEN2-VL-v0.1

Analogous to our method approach with Jina-rerank-m0, we conduct a comprehensive evaluation of MonoQwen2-VL-v0.1 across the full spectrum of task types. Given that MonoQwen2-VL-v0.1 is exclusively trained and tested on the T→VD task, its input configuration is specifically tailored to this particular scenario, as illustrated in Table 13.

Notably, since MonoQwen2-VL-v0.1 does not incorporate additional instructions during training and lacks inherent instruction-following capabilities, we leverage the established T→VD input template to uniformly configure the inputs for all other tasks, as shown under the **Others** in Table 13.

| | Input Format |
|------|--------------|
| **T → VD** | `{doc}\nAssert the relevance of the previous image document to the following`
`query, answer True or False.  The query is:  {query}` |
| **Others** | `{doc}\nAssert the relevance of the previous document to the following`
`query, answer True or False.  The query is:  {query}` |

Table 13: The input template of MonoQwen2-VL-v0.1. **T → VD** is the original input format of it, and we design the input formats for other tasks based on this format, as shown in **Others**.

| Task | Dataset | Query Instruction |
|------|---------|-------------------|
| T→T | ArguAna | Given a claim, find documents that refute the claim. |
| | Climate-FEVER | Given a claim about climate change, retrieve documents that support orrefute the claim. |
| | CQADupStack | Given a question, retrieve detailed question descriptions from Stackexchange that are duplicates to the given question. |
| | DBPedia | Given a query, retrieve relevant entity descriptions from DBPedia. |
| | FiQA2018 | Given a financial question, retrieve user replies that best answer the question. |
| | HotpotQA | Given a multi-hop question, retrieve documents that can help answer the question. |
| | MSMARCO | Given a web search query, retrieve relevant passages that answer the query. |
| | NFCorpus | Given a question, retrieve relevant documents that best answer the question. |
| | NQ | Given a question, retrieve Wikipedia passages that answer the question. |
| | Quora | Given a question, retrieve questions that are semantically equivalentto the given question. |
| | SCIDOCS | Given a scientific paper title, retrieve paper abstracts that are cited bythe given paper. |
| | SciFact | Given a scientific claim, retrieve documents that support or refute theclaim. |
| | Touche2020 | Given a question, retrieve detailed and persuasive arguments that answer the question. |
| | TRECCOVID | Given a query on COVID-19, retrieve documents that answer the query. |
| I→I | Nights | Find a day-to-day image that looks similar to the provided image. |
| T→I | VisualNews | Identify the news-related image in line with the described event. |
| | Fashion200k | Based on the following fashion description, retrieve the best matching image. |
| | Memotion HatefulMemes | Retrieve the meme based on the given caption. |
| T→VD | TAT-DQA ArxivQA DocVQA *MITTI* *WER* | Find a screenshot that relevant to the user's question. |
| I→T | VisualNews | Find a caption for the news in the given photo. |
| | Fashion200k | Find a product description for the fashion item in the image. |
| | GLDv2 | Retrieve the name of the landmark based on the given image. |
| | Memotion HatefulMemes | Retrieve the caption based on the given meme. |
| T→IT | WebQA | Find a Wikipedia image that answers this question. |
| | EDIS | Find a news image that matches the provided caption. |
| IT→T | OVEN | Retrieve a Wikipedia paragraph that provides an answer to the given query about the image. |
| | INFOSEEK | Find a paragraph from Wikipedia that answers my question about this image. |
| | OKVQA | Retrieve documents that provide an answer to the question alongside the image. |
| | VizWiz | Retrieve the correct answer for a question about an image. |
| IT→I | FashionIQ | Find a fashion image that aligns with the reference image and style note. |
| | CIRR | Retrieve a day-to-day image that aligns with the modification instructions of the provided image. |
| IT→IT | OVEN | Retrieve a Wikipedia image-description pair that provides evidence for the question of this image. |
| | INFOSEEK | Find an image and subject description from Wikipedia that answers my question about this image. |
| | E-VQA | Obtain illustrated documents that correspond to the inquiry alongside the provided image. |

Table 14: The instructions for different tasks. We set the instructions for the GME-2B and GMR series models on each task as shown in the Table. *WER* means World Economic Reports, and *MITTI* means MIT Tissue Interaction.

# E MAIN RESULT

## E.1 DETAILED RESULTS

We evaluate all models described in §5 on our benchmark. The evaluation metrics and the detailed results for each dataset are reported in Table 15.

## E.2 THE INFLUENCE OF THE NUMBER OF NEGATIVE

In §5.2, we examine the effect of incorporating negatives in supervised fine-tuning (SFT) and observed that, within the limits of available computational resources, increasing the number of negative examples consistently improved model performance. The best performance was achieved when the number of negative examples reached 16. For comparison, we further conduct experiments on the role of negatives in contrastive learning. As shown in Figure 10, the results indicate that, similar to SFT, a larger number of negative examples leads to better performance. Nevertheless, the overall performance of contrastive learning remains lower than that of supervised fine-tuning.

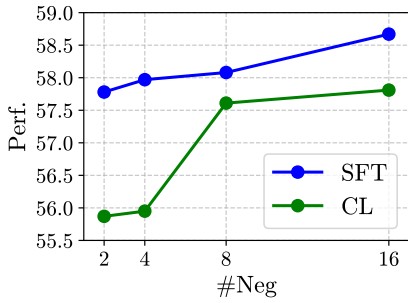

Figure 10: Average performance of the number of negatives per sample.

## E.3 THE INFLUENCE OF THE FROZEN OF LM HEAD

In §4, we observe that SFT can exploit semantic signals from pre-trained token embeddings, whereas CL must learn the score-projection matrix from scratch. To rule out the potential influence of freezing the language modeling (LM) head parameters, we conduct an ablation study on LM head parameter freezing, with the results presented in Table 16. The findings show that freezing or unfreezing the LM head has no effect on SFT. In contrast, CL achieves better performance when the LM head parameters are not frozen. These results suggest that SFT effectively leverages the semantic information embedded in pre-trained token of LLM, while CL requires relearning the score-projection matrix.

|  | *-F* | *-NF* | $\Delta_f$ |
|---|---|---|---|
| SFT | 57.97 | 57.94 | ▼ 0.03 |
| CL | 55.95 | 57.20 | ▲ 1.25 |

Table 16: Impact of frozen of the LM head on performance. *-F* denotes frozen, while *-NF* denotes not frozen.

## E.4 THE INFLUENCE OF NEGATIVE SAMPLE SELECTION

In §5 , we train our models using a 1:1 ratio of randomly sampled negatives and hard negatives. To validate the effectiveness of this negative-sample selection strategy, as well as to examine the sensitivity of SFT and CL to negatives, we conduct an additional analysis in a pure text-based setting. Specifically, we train the models on the MS MARCO using configurations largely consistent with those in §4, and evaluate them on the T→T test sets. The results are presented in the Figure 11.

We observe that both SFT and CL achieve their best performance under the balanced condition, with SFT consistently outperforming CL, demonstrating its superiority. Moreover, SFT exhibits greater robustness to variations in negative-sample selection compared to CL, further highlighting the advantages of the SFT framework.

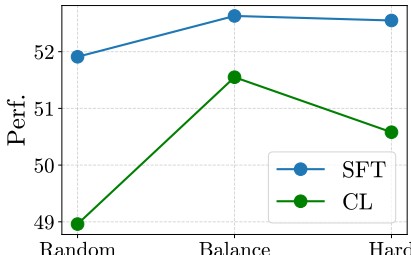

Figure 11: Average performance on the T→T under different negative sampling strategies. *Random* denotes using random negatives, *Balance* denotes a 1:1 mix of random and hard negatives, and *Hard* denotes using hard negatives.

These findings further verify the soundness and accuracy of our negative-sample selection strategy.

| Class | Dataset | Model | | | | | |
|---|---|---|---|---|---|---|---|
| | | **GME-2B** | *Qwen3* | *MonoQwen* | *Jina-m0* | **GMR-3B** | **GMR-7B** |
| T→T (14) | ArguAna† | 47.11 | 86.00 | 50.93 | 56.07 | 80.42 | 84.49 |
| | SCIDOCS† | 22.65 | 26.42 | 18.31 | 22.12 | 25.49 | 28.77 |
| | TRECCOVID† | 79.11 | 87.83 | 79.84 | 85.36 | 87.23 | 85.56 |
| | Quora† | 87.35 | 88.16 | 82.71 | 87.98 | 89.51 | 89.91 |
| | SciFact† | 66.53 | 79.83 | 74.94 | 79.18 | 77.52 | 79.70 |
| | NFCorpus† | 36.90 | 41.88 | 38.29 | 40.99 | 40.51 | 40.81 |
| | Climate-FEVER† | 32.15 | 49.08 | 19.78 | 34.33 | 50.14 | 50.26 |
| | FiQA2018† | 46.35 | 56.25 | 44.11 | 50.72 | 54.79 | 59.64 |
| | HotpotQA† | 70.45 | 82.66 | 71.64 | 80.49 | 82.86 | 83.84 |
| | DBPedia† | 43.17 | 52.69 | 41.75 | 49.60 | 52.99 | 53.96 |
| | Touche2020† | 33.18 | 43.00 | 36.71 | 38.40 | 32.17 | 37.26 |
| | NQ† | 51.22 | 63.33 | 49.08 | 62.06 | 62.49 | 66.48 |
| | MSMARCO† | 40.79 | 44.57 | 35.57 | 43.09 | 45.90 | 47.60 |
| | CQADupStack† | 37.25 | 45.18 | 40.83 | 44.66 | 47.10 | 46.81 |
| | Avg. | 49.59 | 60.49 | 48.89 | 55.36 | 59.22 | 61.08 |
| I→I (1) | Nights⋆ / Avg. | 30.75 | - | 12.59 | 27.50 | 29.76 | 32.83 |
| T→I (4) | Fashion200k* | 25.77 | - | 29.14 | 29.38 | 25.01 | 27.57 |
| | HatefulMemes† | 52.09 | - | 74.93 | 76.57 | 75.07 | 75.19 |
| | Memotion† | 77.41 | - | 93.47 | 93.40 | 93.17 | 93.52 |
| | VisualNews⋆ | 38.55 | - | 37.39 | 38.48 | 42.16 | 48.44 |
| | Avg. | 48.46 | - | 59.46 | 58.73 | 58.85 | 61.18 |
| T→VD (5) | TAT-DQA† | 71.23 | - | 79.99 | 82.05 | 83.23 | 84.00 |
| | DocVQA† | 56.44 | - | 57.51 | 61.69 | 61.48 | 62.87 |
| | ArxivQA† | 84.21 | - | 87.61 | 89.38 | 88.99 | 90.99 |
| | *WER*† | 58.78 | - | 63.00 | 63.47 | 62.13 | 61.00 |
| | *MITTI*† | 61.29 | - | 68.32 | 69.07 | 66.06 | 65.82 |
| | Avg. | 66.39 | - | 71.29 | 73.13 | 72.38 | 72.94 |
| I→T (5) | Fashion200k* | 27.67 | - | 7.55 | 17.14 | 26.22 | 29.80 |
| | HatefulMemes† | 57.85 | - | 32.27 | 80.90 | 81.21 | 81.23 |
| | Memotion† | 80.01 | - | 44.74 | 94.84 | 96.08 | 96.68 |
| | GLDv2† | 59.28 | - | 5.72 | 59.21 | 68.68 | 76.74 |
| | VisualNews⋆ | 38.28 | - | 7.83 | 25.05 | 43.12 | 48.60 |
| | Avg. | 52.62 | - | 19.62 | 55.43 | 63.06 | 66.61 |
| T→IT (2) | WebQA⋆ | 83.03 | - | 87.30 | 87.14 | 86.98 | 87.46 |
| | EDIS⋆ | 71.00 | - | 65.63 | 62.76 | 76.95 | 81.64 |
| | Avg. | 77.02 | - | 76.46 | 74.95 | 81.96 | 84.55 |
| IT→T (4) | OKVQA* | 29.71 | - | 20.13 | 30.34 | 37.71 | 40.09 |
| | VizWiz† | 29.56 | - | 5.11 | 20.36 | 35.96 | 41.29 |
| | INFOSEEK⋆ | 39.77 | - | 23.97 | 36.84 | 59.17 | 63.01 |
| | OVEN⋆ | 60.46 | - | 8.18 | 23.74 | 62.41 | 68.78 |
| | Avg. | 39.88 | - | 14.35 | 27.82 | 48.81 | 53.29 |
| IT→I (2) | FashionIQ* | 26.57 | - | 21.41 | 25.97 | 30.70 | 33.32 |
| | CIRR⋆ | 46.83 | - | 42.09 | 49.33 | 57.24 | 61.46 |
| | Avg. | 36.70 | - | 31.75 | 37.65 | 43.97 | 47.39 |
| IT→IT (3) | INFOSEEK⋆ | 44.61 | - | 35.39 | 53.28 | 73.89 | 76.31 |
| | E-VQA⋆ | 79.11 | - | 55.81 | 61.21 | 84.66 | 86.08 |
| | OVEN⋆ | 76.96 | - | 16.28 | 40.12 | 78.68 | 84.17 |
| | Avg. | 66.89 | - | 35.83 | 51.54 | 79.08 | 82.19 |
| ALL(40) | Avg. | 52.54 | - | 44.20 | 54.36 | 61.40 | 63.85 |

Table 15: Detailed scores of each model on various datasets on *MRB*. *WER* denotes World Economic Reports, and *MITTI* refers to MIT Tissue Interaction. For datasets marked with ⋆/*, we report Recall@5/Recall@10, and NDCG@5 / NDCG@10 is used for †-labeled / †-labeled datasets.

## F  THE USE OF LARGE LANGUAGE MODELS (LLMS)

We primarily employ large language models (LLMs) for manuscript refinement, encompassing editing and stylistic enhancement to improve clarity, coherence, and consistency. Furthermore, our GMR models are built upon the Qwen2.5-VL-Instruction series as the foundational architecture, leveraging its multimodal understanding and instruction-following capabilities, while incorporating task-specific supervised fine-tuning to achieve high-performance reranking across diverse scenarios.

## G  LIMITATION

In this work, we introduce MRB, a benchmark designed for training and evaluating multimodal reranking tasks. To address this challenge, we investigate strategies for adopting Multimodal Large Language Models (MLLMs) into general-purpose multimodal reranking models, and propose GMR, a reranking model capable of handling candidates across different modalities. Despite these contributions, our work has the following limitations:

• Single-language constraint. Although the backbone model, Qwen2.5-VL-Instruction, supports multiple languages, we trained and evaluated GMR exclusively in English. Consequently, the performance of GMR in other languages remains unexplored.

• Single-image constraint for queries and documents. For reasons of training efficiency and limited availability of relevant data, both queries and candidates in MRB are restricted to a single image for each query and document. As a result, the benchmark cannot assess performance on interleaved inputs that involve multiple images and texts.

