# OpenReview forum: "Supervised Fine-Tuning or Contrastive Learning? Towards Better Multimodal LLM Reranking"
_ICLR.cc/2026/Conference — ICLR 2026 Poster_

### Official Review · Reviewer_xxqB · 2025-10-31

**Soundness:** 2
**Presentation:** 3
**Contribution:** 3
**Rating:** 6
**Confidence:** 3

**Summary:**

This paper compares two popular approaches for training reranking models—Supervised Fine-Tuning (SFT) and Contrastive Learning (CL)—in the context of large language model (LLM)-based reranking. The paper introduces a unified framework that decomposes the loss functions into two key components: weight and direction. The authors conduct a comprehensive theoretical analysis and empirical comparison between SFT and CL using the Universal Multimodal Retrieval (UMR) task. Their findings suggest that SFT offers better performance than CL due to its stronger optimization signals, specifically in the weight component. The authors also introduce a new multimodal reranking benchmark (MRB), which consists of 40 datasets across various modalities. Through this analysis, they develop new state-of-the-art rerankers, GMR-3B and GMR-7B, achieving superior results in the MRB benchmark.

**Strengths:**

1. The paper introduces a novel framework to analyze two popular reranking training methods, SFT and CL, by decomposing their loss functions. This decomposition into weight and direction provides new insights into their effectiveness. Additionally, the creation of the MRB benchmark is a significant contribution to the multimodal reranking field, offering a diverse and comprehensive dataset for evaluating models.

2. The authors provide a thorough theoretical analysis combined with extensive experiments. They rigorously evaluate both SFT and CL, demonstrating that SFT outperforms CL in terms of model performance. The experiments are well-designed, and the results are statistically significant.

3. The paper is clearly written, with well-defined objectives, methodology, and results. The use of equations and illustrations, such as Figure 1, helps clarify the distinctions between the two training paradigms. The explanations of the framework, loss function decomposition, and the MRB benchmark are easy to follow.

4. The findings that SFT offers superior performance to CL for LLM-based reranking are crucial for future research and practical applications in multimodal information retrieval. The introduction of the MRB benchmark and the state-of-the-art models developed (GMR-3B and GMR-7B) have the potential to impact a wide range of tasks in multimodal retrieval systems.

**Weaknesses:**

1. While the MRB benchmark is comprehensive, it might be seen as overly focused on tasks where image-text retrieval is central. Future work could explore additional modalities or larger-scale datasets to evaluate the generalizability of the reranking models to other forms of multimodal data, such as video or audio.

2. Given the reliance on MRB for validation, there is a risk of overfitting to this specific benchmark, as many models may perform well only on the datasets included in MRB but struggle on entirely new or more complex multimodal benchmarks. The paper could benefit from additional cross-benchmark evaluations to test the robustness of the proposed models.

**Questions:**

1. Since the MRB benchmark is restricted to single-image inputs, would the performance of the models differ when dealing with queries or documents containing multiple images or mixed modalities (e.g., a combination of images and text)?

2. The paper mentions the use of random and hard negative sampling strategies. How do the negative sampling strategies influence the performance in specific modalities (e.g., images vs. text)? Is there a particular strategy that consistently outperforms the others across all tasks?

3. While the paper primarily focuses on SFT and CL, would it be worth exploring hybrid architectures that combine the strengths of both methods? How could a mixed approach impact the performance of multimodal rerankers?

---

> ### Author Response · Authors · 2025-11-21
> **Reply to Weakness 1-2, Question 1**
>
> We sincerely thank you for the valuable comments and insightful suggestions. In the following, we provide detailed responses to each point.
>
> ---
>
> # Reply to Weakness 1
> > While the MRB benchmark is comprehensive, it might be seen as overly focused on tasks where image-text retrieval is central. Future work could explore additional modalities or larger-scale datasets to evaluate the generalizability of the reranking models to other forms of multimodal data, such as video or audio.
>
> Thank you for the thoughtful suggestion. Prior multimodal retrieval benchmarks—such as [MM-Embed](https://arxiv.org/abs/2411.02571)[1] and [GME](https://arxiv.org/abs/2412.16855)[2] —have also focused primarily on image–text retrieval, and our work follows this widely adopted evaluation setting to ensure fair comparison with existing methods.
>
> That said, we fully agree that exploring a broader spectrum of modalities would further strengthen the understanding of a reranker’s generalizability. In future work, we plan to incorporate larger-scale and more diverse multimodal datasets, including video–text and audio–text retrieval benchmarks, to more comprehensively evaluate the cross-modal robustness of our reranking model.
>
> [1] MM-Embed: Universal Multimodal Retrieval with Multimodal LLMs, ICLR, 2025
>
> [2] GME: Improving Universal Multimodal Retrieval by Multimodal LLMs, CVPR, 2025
>
> ---
>
> # Reply to Weakness 2
> > Given the reliance on MRB for validation, there is a risk of overfitting to this specific benchmark, as many models may perform well only on the datasets included in MRB but struggle on entirely new or more complex multimodal benchmarks. The paper could benefit from additional cross-benchmark evaluations to test the robustness of the proposed models.
>
> Thank you for raising this important point. In MRB, we deliberately include **40** test sets across **9** task categories (refer to Section 5) to provide broad and diverse evaluation coverage for general multimodal retrieval. We believe this scale substantially reduces the risk of overfitting to any specific dataset.
>
> For example, in the text–text retrieval setting, although MSMARCO is used for training, our evaluation spans **14** BEIR datasets, ensuring that the model is tested on distributions it has never seen during training.
>
> That said, we agree that broader cross-benchmark validation would further strengthen the assessment of model robustness. In future work, we plan to incorporate evaluations on additional modalities—such as video and audio—to more comprehensively examine the generalization capability of the proposed reranker.
>
> ---
>
> # Reply to Question 1
> >Since the MRB benchmark is restricted to single-image inputs, would the performance of the models differ when dealing with queries or documents containing multiple images or mixed modalities (e.g., a combination of images and text)?
>
> Thank you for your question. Regarding inputs that include multiple images:
>
> 1. The reranking setting of it2it naturally supports multi-image inputs, as both queries and documents may contain image.
>
> 2. The underlying Qwen2.5-VL model supports dynamic visual token allocation, meaning that the number of visual tokens is adaptively determined by image resolution. This allows us to concatenate multiple images into a single composite image while preserving the single-image input format used in MRB. We therefore do not expect significant performance degradation when extending to multi-image or mixed-modality cases.
>
> We are currently making active efforts to expand our evaluation, including adding multi-image benchmarks (e.g., [MRMR](https://arxiv.org/pdf/2510.09510)[1]) and extending the dataset to better cover multi-image scenarios.
>
> [1] MRMR: A REALISTIC AND EXPERT-LEVEL MUL-TIDISCIPLINARY BENCHMARK FOR REASONING-INTENSIVE MULTIMODAL RETRIEVAL, Arxiv, 2025

---

> ### Author Response · Authors · 2025-11-21
> **Reply to Question 2-3**
>
> ---
> # Reply to Question 2
> > The paper mentions the use of random and hard negative sampling strategies. How do the negative sampling strategies influence the performance in specific modalities (e.g., images vs. text)? Is there a particular strategy that consistently outperforms the others across all tasks?
>
> Thank you for the question. For negative sampling, we follow prior work and adopt a 1:1 ratio of random and hard negatives during training, as suggested in prior studies (e.g., [mGTE](https://arxiv.org/abs/2407.19669)[1]) on negative sampling in reranking. For hard negative mining, we follow strategies commonly used in multimodal embedding models (e.g., [GME](https://arxiv.org/abs/2412.16855)[2]), which directly extend to our reranking setting.
>
> Importantly, we consider this sampling strategy to be modality-agnostic: the balance between random and hard negatives is expected to have a similar impact across different modalities.
>
> To verify our setting, we additionally conduct experiments on a pure text retrieval task (MS MARCO), using the same SFT/CL training settings as in the main paper, and evaluate in the t2t test datasets. We compared three strategies: Random-only, Balanced (1:1), and Hard-only negative sampling. The results are shown below:
>
> |Model|	Random	| Balance |	Hard |
> |-----|:---------:|:---------:|:------:|
> |SFT  | 51.91	| 52.63	  |52.55 |
> |CL	  | 48.96   | 51.55	  |50.58 |
>
> The balanced 1:1 sampling strategy consistently provides the best performance, further supporting its effectiveness and generalizability.
>
> [1] mGTE: Generalized Long-Context Text Representation and Reranking Models for Multilingual Text Retrieval, EMNLP, 2024
>
> [2] GME: Improving Universal Multimodal Retrieval by Multimodal LLMs, CVPR, 2025
>
> ---
> # Reply to Question 3
> > While the paper primarily focuses on SFT and CL, would it be worth exploring hybrid architectures that combine the strengths of both methods? How could a mixed approach impact the performance of multimodal rerankers?
>
> We agree that exploring hybrid architectures that combine the strengths of both SFT and CL is indeed valuable. For example, one could simultaneously train SFT using the yes/no LM head while training CL with a randomly initialized MLP, allowing the two supervision signals to complement each other and employ dual-checking strategies to enhance ranking consistency. Another direction would be to design composite weighting mechanisms that integrate both SFT and CL objectives. Such hybrid approaches may further improve robustness against hard negatives, strengthen cross-modal alignment, and ultimately enhance the overall performance of multimodal rerankers. We consider this a promising direction and plan to investigate it in future work.

---

> ### Author Response · Authors · 2025-11-26
>
> Dear Reviewer xxqB：
>
> We have revised our paper and highlighted all changes in blue. In Section 5.3, we include additional experiments in the full-modality, multi-image input scenario. By evaluating on the Knowledge subset of MRMR, we show that our model also delivers strong performance in multi-image settings and exhibits a consistent upward trend when increasing the reranking depth from top-25 to top-100, underscoring its robustness. In Appendix E.4, we add comparative experiments and analysis on different negative-sample selection strategies in the text-only setting.  We hope these additional experiments address your concerns. We sincerely appreciate your insightful feedback and look forward to your response and further discussion.

---

### Official Review · Reviewer_W9sD · 2025-10-31

**Soundness:** 3
**Presentation:** 3
**Contribution:** 2
**Rating:** 6
**Confidence:** 3

**Summary:**

This paper asks whether Supervised Fine-Tuning (SFT) or Contrastive Learning (CL) is inherently better for LLM-based (multimodal) reranking. Authors build a unified reranking loss (URL) that decomposes training signals into two orthogonal parts—weight (magnitude) and direction (update vector)—such that SFT and CL can be compared under the same lens. Empirically, SFT beats CL (on a newly assembled, broad Multimodal Reranking Benchmark). Ablating URL shows the weighting scheme explains most of the gap while update direction matters less. Finally, the authors train GMR-3B/7B rerankers with SFT and report SOTA on MRB.

**Strengths:**

* Multi-modal retrievers are very popular these days, and less work is dedicated to multi-modal rerankers. It is an interesting topic per se—given how rerankers usually boost performance in any given setting—we can see how GMR (but also the tested baselines) improve over the tested retriever (GME-2B).
* Overall the paper is clear, straight forward and well written (a few things could be improved though, see later). Related works contain most of the relevant references I am aware of. I would maybe add [1] somewhere (the first paper doing a cross-encoder with BERT).
* The weight/direction derivation gives a mechanistic account of why SFT wins: SFT’s weights are per-document (no across-negative normalization), while CL’s weights couple all negatives, which weakens positive signal magnitude-especially when many easy negatives are present. The derivation looks correct to me. The loss decomposition shows that supervised fine-tuning (SFT) and contrastive learning (CL) differ mainly in how they weigh each training example, not in the basic direction of the update.
* This feels like an interesting (**although not ground breaking**) contribution imo. But the main takeaway is SFT > CL for training pointwise rerankers, which is a valuable finding by itself.
* After ablations (SFT vs CL), authors train two multi-modal rerankers (GMR 2B and GMR 7B) on 1.5M instances. They also assemble a new benchmark MRB (40 **existing** datasets; multiple modalities and tasks), making results quite robust; Table 4 shows GMR outperforming strong baselines across categories. Note that the baselines are 2B (or 4B for Qwen).

[1] Passage Re-ranking with BERT

**Weaknesses:**

* Novelty/contribution feels a bit limited, but I still could see this paper fit into ICLR (especially if authors release the code and models as stated in the paper). Especially, it focuses (by design) on pointwise approaches, and obfuscate listwise rerankers which might become more predominant in the future (due to the nature of LLMs).
* “MRB excludes datasets where the retriever already scores very high”; in these cases rerankers have no effect? Scores could be very high for retrievers but still be improved by rerankers in practice no? Anyway, authors could also have included “weaker” retrievers; this would even show more the effect of reranking. Maybe varying topk could have been beneficial too, to test whether SFT’s advantage holds when the negative mix changes.
* Although the paper is clear overall, some parts could be improved/clarified (not critical). There are many typos/weird formulations across the paper; for instance, in the Supervised Fine-Tuning paragraph (Section 3.2), “The objective is predicting correct next token” or “Then predict the likelihood of “yes” [...]”. And there are many more...
* It seems that what makes ICL different from SFT is the fact that ICL couples all negatives through a softmax denominator (vs independently for SFT); it feels that the impact of negatives could have been studied further. Do findings still apply with 1 neg? 100 negs? What about temperature which is barely discussed (although is it common practice)?

**Questions:**

See other questions in weaknesses; see also

* In Appendix E.3, unfreezing the LM head benefits CL but not SFT. Could you explore whether adding a small projection layer (instead of the LM head) narrows the gap? That would indicate whether SFT’s advantage comes from leveraging pretrained lexical priors?
* Do you think your analysis could extend to pairwise or listwise rerankers where the loss itself defines relative weighting across candidates rather than independent pairs? Clarifying this could help generalize the decomposition beyond pointwise reranking.
* Authors directly tackle multi-modal reranking; could you show the same findings in standard text scenarios (mono- and/or multi-lingual)?

---

> ### Author Response · Authors · 2025-11-21
> **Reply to Weakness 1-3**
>
> We are grateful for your time and thoughtful review. Our point-by-point replies to the concerns are presented as follows.
>
> ---
>
> # Reply to Weakness 1
> > Novelty/contribution feels a bit limited, but I still could see this paper fit into ICLR (especially if authors release the code and models as stated in the paper). Especially, it focuses (by design) on pointwise approaches, and obfuscate listwise rerankers which might become more predominant in the future (due to the nature of LLMs).
>
> Thank you very much for your positive assessment and constructive feedback. As stated in the paper, all datasets used in our work are sourced from the open-source community, and we have included the training and evaluation code in the supplementary materials. We also plan to release our trained model to further support research progress in multimodal reranking.
>
> We fully agree that listwise reranking approaches may become increasingly important as large language models continue to evolve. While our current work focuses primarily on pointwise methods by design, we view listwise rerankers as a valuable and promising direction, and we plan to explore them in future extensions of this work.
>
> ---
>
> # Reply to Weakness 2
> > “MRB excludes datasets where the retriever already scores very high”; in these cases rerankers have no effect? Scores could be very high for retrievers but still be improved by rerankers in practice no? Anyway, authors could also have included “weaker” retrievers; this would even show more the effect of reranking. Maybe varying topk could have been beneficial too, to test whether SFT’s advantage holds when the negative mix changes.
>
> Thank you for the thoughtful comments. Our clarifications are as follows:
>
> 1. On excluding near-saturated datasets.
> We excluded datasets where retriever performance is already extremely high because they provide limited discriminatory power and may even produce misleading conclusions. For instance, on Flickr30k, GME-2B scores 99.0 while GME-7B scores 98.9, suggesting that (i) these datasets are too simple to reveal reranker differences, and (ii) label noise may cause stronger models to score lower. Thus, MRB focuses on datasets that more reliably differentiate rerankers.
>
> 2. On including weaker retrievers.
> We agree this would further highlight the benefits of reranking. We are currently acquiring resources and plan to add weaker retrievers (e.g., [VISTA](https://arxiv.org/abs/2406.04292)[1]).
>
> 3. On varying top-k.
> We conduct additional analyses with different top-k settings. Increasing top-k (Top-25 → Top-100) amplifies the gap between SFT and CL, indicating that SFT remains more robust under harder negative mixes.
> | Model | Top25 |	Top100 |  Δ |
> |-------|:-------:|:----------:|:-----------:|
> | SFT   | 58.09 |	58.88  |	+0.79  |
> | CL    | 56.40 |	56.76  |	+0.36  |
>
>
> [1] VISTA: Visualized Text Embedding For Universal Multi-Modal Retrieval, ACL, 2024
>
> ---
>
> # Reply to Weakness 3
> > Although the paper is clear overall, some parts could be improved/clarified (not critical). There are many typos/weird formulations across the paper; for instance, in the Supervised Fine-Tuning paragraph (Section 3.2), “The objective is predicting correct next token” or “Then predict the likelihood of “yes” [...]”. And there are many more...
>
> Thank you again for your careful reading of our paper. We apologize for any confusion these inaccuracies may have caused, and we are committed to improving the clarity and overall writing quality in the revised version.

---

> ### Author Response · Authors · 2025-11-21
> **Reply to Weakness 4, Question 1**
>
> ---
> # Reply to Weakness 4
> > It seems that what makes ICL different from SFT is the fact that ICL couples all negatives through a softmax denominator (vs independently for SFT); it feels that the impact of negatives could have been studied further. Do findings still apply with 1 neg? 100 negs? What about temperature which is barely discussed (although is it common practice)?
>
> Thank you for raising this insightful question. Our responses are as follows:
>
> 1. On the impact of negative samples.
> As discussed in Appendix C.3, we adopt a 1:1 ratio of random and hard negatives, which sets the minimum number of negatives to 2. In practice, rerankers typically do not use very large numbers of negatives due to their computational cost and task formulation. For example, the text reranker [GTE-multilingual-reranker-base](https://arxiv.org/abs/2407.19669)[1] uses only 10 negatives. Following this convention, we study the effect of 2–16 negatives and observe consistent trends for both SFT and CL. This range aligns with common reranker training settings; therefore, we did not extend the analysis to much larger negative counts.
>
> 2. On the temperature parameter.
> We did not include temperature tuning in our study because contrastive training for point-wise rerankers generally does not require a temperature parameter, unlike embedding-based models. This practice follows prior work such as: [FlagEmbedding](https://github.com/FlagOpen/FlagEmbedding/blob/master/FlagEmbedding/abc/finetune/reranker/AbsModeling.py)[2], [Distillation versus Contrastive Learning](https://arxiv.org/pdf/2507.08336v3)[3] and [GTE-multilingual-reranker-base](https://arxiv.org/abs/2407.19669)[1].
> The key reason is that:
> Embedding models compute similarity via cosine similarity in [0, 1], where temperature is required to enlarge the positive–negative margin.
> Pointwise rerankers, however, use the raw LM head score (e.g., “yes” logits), which already spans a much larger dynamic range (e.g. 0–50). This naturally provides sufficient separation between positives and negatives, making temperature unnecessary—and applying it could even shrink gradients and hinder training.
> We acknowledge that our earlier explanation may have been unclear and will revise the manuscript to reflect this distinction more clearly.
>
> [1] mGTE: Generalized Long-Context Text Representation and Reranking Models for Multilingual Text Retrieval, EMNLP, 2024
>
> [2] BGE M3-Embedding: Multi-Lingual, Multi-Functionality, Multi-Granularity Text Embeddings Through Self-Knowledge Distillation, ACL, 2024
>
> [3] Distillation versus Contrastive Learning: How to Train Your Rerankers, Arxiv, 2025
>
> ---
> # Reply to Question 1
> > In Appendix E.3, unfreezing the LM head benefits CL but not SFT. Could you explore whether adding a small projection layer (instead of the LM head) narrows the gap? That would indicate whether SFT’s advantage comes from leveraging pretrained lexical priors?
>
> Thank you for the insightful suggestion. Our thinking aligns closely with yours. After observing in Appendix E.3 that unfreezing the LM head benefits CL but not SFT, we also hypothesized that the advantage of SFT might stem from leveraging pretrained lexical priors. To verify this, we conducted additional experiments by introducing a small, randomly initialized projection layer in place of the LM head. The results are reported in Section 4.2 under “Is it possible to learn a better direction?”. We found that adding this projection layer improves the performance of CL but leads to a performance drop for SFT, and under this setting CL surpasses SFT. (Naturally, the best overall performance is still achieved by SFT when using the pretrained yes/no LM head.) These findings support the conclusion that part of SFT’s advantage indeed comes from benefiting from the pretrained lexical priors encoded in the LM head.

---

> ### Author Response · Authors · 2025-11-21
> **Reply to Question 2-3**
>
> ---
>
> # Reply to Question 2:
> > Do you think your analysis could extend to pairwise or listwise rerankers where the loss itself defines relative weighting across candidates rather than independent pairs? Clarifying this could help generalize the decomposition beyond pointwise reranking.
>
> Thank you for the insightful question. We believe parts of our analysis naturally extend to pairwise reranking. A pairwise setup is often formulated as **{query; doc₁; doc₂; “Is doc₁ more relevant than doc₂?” → yes/no}** or **{query; docA; docB;“Which one is more relevant?” → A/B}**.
>
> In this case, applying SFT is structurally similar to the pointwise formulation: when doc₁ is the preferred (positive) document, the instance is treated as a positive example, and when doc₁ is the less relevant (negative) document, the instance is treated as a negative example. Based on our current analysis, we expect that directly leveraging the model’s native tokens (such as **yes/no** or **A/B**)—rather than introducing a newly initialized MLP head—would likely yield better performance.
>
> For listwise reranking, the scenario is more complex. The task can be framed as **{query; doc₁, doc₂, …, docₙ → generate a ranked order}**, and the loss function differs substantially from the pointwise decomposition we analyze. We are actively investigating how our framework might generalize to this setting. If you have additional ideas or perspectives, we would be very happy to discuss them further.
>
> ---
>
> # Reply to Question 3:
> > Authors directly tackle multi-modal reranking; could you show the same findings in standard text scenarios (mono- and/or multi-lingual)?
>
> Yes, we also conduct experiments in a text-only setting to verify whether our conclusions generalize beyond multimodal data. Specifically, we train both SFT and CL on MSMARCO and evaluate them on the T$\rightarrow$T test datasets. The results show that SFT consistently outperforms CL, demonstrating that our findings hold in standard textual reranking scenarios as well.
>
> |Model| Avg(T$\rightarrow$T) |
> |-----|:---------:|
> |SFT  | 52.63	|
> |CL	  | 51.55	|

---

> ### Author Response · Authors · 2025-11-26
>
> Dear Reviewer W9sD：
>
> We have revised our paper and highlighted all changes in blue. In Section 3, we clarify several statements that were previously ambiguous. In Section 5.2, we add a comparative analysis of model performance under different top-K settings. In Appendix E.4, we include experiments on negative-sample selection strategies in the text-only scenario, along with an analysis of the relative performance of SFT and CL in standard text-based settings. We hope that these additional experiments address your concerns and that our revisions resolve any misunderstandings caused by unclear wording in the original manuscript. We sincerely appreciate your valuable feedback and look forward to your response and further discussion.

---

> ### Comment · Reviewer_W9sD · 2025-11-26
>
> Dear authors,
> Thank you for your detailed response, which has addressed most of my concerns and clarified several points of confusion.
>
> After careful consideration, I am inclined to maintain my original score of 6. While this is a solid contribution, it remains somewhat limited in terms of novelty and findings, as I noted in my initial review.

---

### Official Review · Reviewer_XjCJ · 2025-10-31

**Soundness:** 3
**Presentation:** 3
**Contribution:** 2
**Rating:** 4
**Confidence:** 4

**Summary:**

This paper systematically investigates two primary training paradigms for large language model (LLM)-based reranking—Supervised Fine-Tuning (SFT) and Contrastive Learning (CL)—within a universal multimodal retrieval setting. The authors introduce a unified theoretical framework that decomposes both objectives into weight and direction components, enabling a fair analytical comparison. Through extensive probing experiments, they show that the weighting mechanism in SFT provides stronger optimization signals than CL, accounting for most of the performance gap. The study culminates in the development of GMR (General Multimodal Reranker) models (3B and 7B parameters), trained via SFT, that achieve state-of-the-art results on a newly constructed MRB benchmark comprising 40 multimodal retrieval datasets. The work provides both theoretical insight and practical advances for multimodal reranking.

**Strengths:**

1, Extensive experiments, including ablations, controlled comparisons, and probing analyses, convincingly support the claims.

2, MRB provides a comprehensive, standardized testbed likely to benefit the broader research community.

3, GMR models consistently outperform existing multimodal rerankers across 40 datasets.

4, The paper addresses a timely and underexplored question—the objective design for LLM-based reranking—bridging theoretical understanding with real-world performance.

**Weaknesses:**

1, The conclusion of this theoretical analysis – "CL computes the weight using all positive and negative documents within a sample, while SFT assigns weights independently per document, making this the likely key factor in performance variation" – is extremely obvious. It can be verified simply by checking which terms are involved in the calculation when computing the loss. What we are more curious about is:  Why would training with both positive and negative examples lead to better or worse performance? We need a theoretical explanation from perspectives such as generalization, embedding space, and the difficulty of training convergence. These are exactly the areas where mathematical analysis should play its role.

2, In the ai search era, we mainly care about llm reranker for text. Experiments on mllm rerankers are somehow limited.

3, While weight and direction are well formalized, their intuitive interpretation in semantic space could be further elaborated. e.g. what's the difference between "generating" the score and the traditional "encoding then scoring with mlp" pipeline?

**Questions:**

1, What about list-wise reranking (where llm generates a sequence of doc ids)? e.g (https://aclanthology.org/2024.emnlp-main.491.pdf) In this case, what's the difference between sft and cl?

2,  How sensitive is the performance gap between SFT and CL to negative sampling strategies or batch size?

3, Does SFT’s superior performance persist when the LLM backbone is not generative? e.g. A randomly initialized LLM

---

> ### Author Response · Authors · 2025-11-21
> **Reply to Weakness 1-3**
>
> We sincerely thank you for the valuable comments and insightful suggestions. In the following, we provide detailed responses to each point.
>
> ---
>
> # Reply to W1
>
> > The conclusion of this theoretical analysis – "CL computes the weight using all positive and negative documents within a sample, while SFT assigns weights independently per document, making this the likely key factor in performance variation" – is extremely obvious. It can be verified simply by checking which terms are involved in the calculation when computing the loss. What we are more curious about is: Why would training with both positive and negative examples lead to better or worse performance? We need a theoretical explanation from perspectives such as generalization, embedding space, and the difficulty of training convergence. These are exactly the areas where mathematical analysis should play its role.
>
> We acknowledge your point and will continue to investigate these perspectives in our follow-up work. If you have further ideas or suggestions, we would be very happy to discuss them and explore potential directions together.
>
> ---
>
> # Reply to W2
>
> > In the ai search era, we mainly care about llm reranker for text. Experiments on mllm rerankers are somehow limited.
>
> Indeed, text-based LLM rerankers have received more attention and research focus due to their broad market demand and wide range of application scenarios. This is also the main reason why, while building a general reranking framework, we placed particular emphasis on text reranking tasks. Statistically, in our training set, text retrieval (t2t) tasks account for 32.1% of the nine task categories (refer to Appendix C.1), and 35% (14 out of 40) in the test set. Therefore, we have placed great importance on evaluating the model’s performance in text-based tasks throughout training and testing.
>
> To more comprehensively verify the effectiveness of our model, we selected [Qwen3-Reranker](https://arxiv.org/abs/2506.05176)[1], one of the current leading text reranking models, as a comparison baseline to demonstrate that our general multimodal reranker remains competitive even in text-only scenarios. Nevertheless, we also believe that multimodal reranking research possesses stronger generalization potential and long-term value.
>
> Besides, we also conduct experiments in a text-only setting to verify whether our conclusions generalize beyond multimodal data. Specifically, we train both SFT and CL on MSMARCO and evaluate them on the T$\rightarrow$T test datasets. The results show that SFT consistently outperforms CL, demonstrating that our findings hold in standard textual reranking scenarios as well.
>
> |Model| Avg(T$\rightarrow$T) |
> |-----|:---------:|
> |SFT  | 52.63	|
> |CL	  | 51.55	|
>
> [1] Qwen3 Embedding: Advancing Text Embedding and Reranking Through Foundation Models, Arxiv, 2025
>
> # Reply to W3
>
> > While weight and direction are well formalized, their intuitive interpretation in semantic space could be further elaborated. e.g. what's the difference between "generating" the score and the traditional "encoding then scoring with mlp" pipeline?
>
> Conceptually, the key difference lies in how the model leverages pretrained semantic priors:
>
> Score generation (our approach) directly uses the LM head to produce a relevance score, allowing the model to fully exploit the pretrained lexical and semantic knowledge embedded in the LLM. The scoring direction is therefore aligned with the model’s intrinsic semantic space.
>
> In contrast, the encode → MLP scoring pipeline introduces an additional MLP that must be trained from scratch, requiring the model to relearn a scoring function that is not naturally aligned with the pretrained LM head. This can lead to weaker generalization.
>
> In short, “generating” the score leverages the LLM’s pretrained semantic space, whereas MLP scoring learns a new semantic projection, which may be less efficient and less effective.

---

> ### Author Response · Authors · 2025-11-21
> **Reply to Question 1-3**
>
> ---
>
> # Reply to Question 1
>
> > What about list-wise reranking (where llm generates a sequence of doc ids)? e.g (https://aclanthology.org/2024.emnlp-main.491.pdf) In this case, what's the difference between sft and cl?
>
> Thank you for raising this point. List-wise reranking methods train the model to directly generate an ordered sequence of document IDs. Unlike point-wise reranking, this formulation no longer relies on explicit positive/negative samples. Instead, the model learns to produce an entire ranking as a structured output.
>
> Because the supervision signal is sequence-level rather than document-level, the distinction between SFT and CL becomes fundamentally different from the point-wise case. SFT remains a natural fit for next-token prediction over document-ID sequences, whereas defining contrastive objectives in this structured listwise space is considerably more complex and not directly analogous to point-wise CL.
>
> Our current work focuses on point-wise reranking, and extending the analysis to list-wise settings requires additional modeling of sequence-level dependencies and ranking-level supervision. We are actively exploring this direction as part of future work. If you have further ideas or suggestions, we would be very glad to discuss them.
>
> ---
>
> # Reply to Question 2
>
> > How sensitive is the performance gap between SFT and CL to negative sampling strategies or batch size?
>
> Thank you for the question. We additionally conducted experiments on a pure text retrieval task (MS MARCO), using nearly the same SFT/CL training configurations as in the main paper, and evaluated on the t2t test sets. We compared three negative sampling strategies—Random-only, Balanced (1:1), and Hard-only. The results are shown below:
>
> |Model|	Random	| Balance |	Hard |
> |-----|:---------:|:---------:|:------:|
> |SFT  | 51.91	| 52.63	  |52.55 |
> |CL	  | 48.96   | 51.55	  |50.58 |
>
> Across all settings, the balanced sampling strategy consistently yields the best performance, demonstrating its effectiveness and generalizability. We also observe that CL is more sensitive to the choice of negative sampling, whereas SFT exhibits stronger robustness and consistently outperforms CL, further highlighting the advantage of SFT in reranking.
>
> Regarding batch size, we expect SFT and CL to be relatively insensitive to batch size in reranking setups, and we are actively securing additional computational resources to further validate this empirically.
>
> ---
>
> # Reply to Question 3
>
> >  Does SFT’s superior performance persist when the LLM backbone is not generative? e.g. A randomly initialized LLM
>
> When the backbone is randomly initialized, the model lacks any prior knowledge. Under this setting, we expect SFT and CL to exhibit similar performance, as neither method can leverage pretrained semantic representations. We are actively seeking computational resources to conduct a experimental verification.

---

> > ### Comment · Reviewer_XjCJ · 2025-11-23
> >
> > Thank you for your response, which has resolved some of my confusion. I believe there is still room for further improvement and refinement in this work, and I am pleased that you are open to these suggestions. I have ultimately decided to maintain my original rating.

---

> > > ### Author Response · Authors · 2025-11-27
> > >
> > > Dear Reviewer XjCJ:
> > >
> > > We have conducted additional experiments to analyze the effect of batch size in the text-only setting. Specifically, we train models using batch sizes(BS) of 96, 144, and 192 instances per batch, where each instance contains one positive sample and four negatives. As shown in the table below, the results indicate that, compared with the influence of negative-sample quantity and selection strategies, the reranker is relatively insensitive to batch-size variation. Both SFT and CL exhibit only minor performance fluctuations.
> > > | Model | BS-96    | BS-144   | BS-192   |
> > > | ---------- | ----- | ----- | ----- |
> > > | SFT    | 53.02 | 52.78 | 52.63 |
> > > | CL     | 51.71 | 52.05 | 51.55 |
> > >
> > > We also conduct experiments on a randomly initialized model using Qwen2.5-0.5B-Instruction. We observe that training from a fully random initialization leads to extremely slow convergence, making it difficult to obtain a fair and meaningful comparison between SFT and CL under comparable conditions. Therefore, we believe that results obtained under this setting may not provide sufficiently informative or representative evidence for methodological comparison.
> > >
> > > In addition, we have revised our paper and highlighted all changes in blue. In Appendix E.4, we add comparative experiments and analysis on different negative-sample selection strategies in the text-only setting.
> > >
> > > Finally, we sincerely appreciate your thoughtful feedback and hope that these additional results address your concerns. If any issues remain unresolved, or if your suggestions regarding further improvement and refinement refer to specific aspects that require additional analysis, we would be grateful if you could kindly clarify them. We believe that further discussion will help us strengthen the paper, and we remain fully committed to making changes for clarification and to address your concerns.

---

### Official Review · Reviewer_51SZ · 2025-11-03

**Soundness:** 3
**Presentation:** 3
**Contribution:** 3
**Rating:** 8
**Confidence:** 5

**Summary:**

This paper explores the tradeoffs between direct supervision (binary cross entropy) and contrastive loss (InfoNCE) for training reranking models. They decompose each loss into two parts: weight (gradient magnitude) and direction. Given that both losses rely on the same data, one can mix and match components to have a new loss that uses the weight of direct supervision and direction of contrastive (and vice versa). The authors find that direct supervision is more effective than contrastive in their experiments, and analysis reveals that it is in large part because of the "weight" component of direct supervision --- extrapolating from the analysis, my sense is the weight is potentially more robust against easy negatives, given that contrastive leads to relatively small gradients. They train a reranker with direct supervision and see strong results across multiple benchmarks.

**Strengths:**

S1. The paper nicely walks through the differences of direct supervision and contrastive loss, fundamental concepts for embedding and reranker training.

S2. There was extensive analysis of the two losses, including of their hybrid forms. This analysis reveals some insights for reranker training, although it was unclear how this influenced their main result experiments. Those experiments simply could have use direct supervision and contrastive as a hyperparameter to explore.

S3. The authors train multiple versions of Qwen2.5-VL-Instruction for reranking with strong performance across multiple benchmarks. Details for training and data selection are provided. The authors also say they will release code, data and models.

**Weaknesses:**

W1. The paper is not self contained in the main text. Many of the main results are in the appendix. For this statement "Rival and surpass leading textual reranker", there is no easy way for the reader to verify. The relevant results are in the appendix, and not averaged across datasets.

W2. There are only experiments when reranking the top-100 retrieved documents, but many real world settings will require reranking more or less than this amount. Given that the reranker here is pointwise, it would be relatively straightforward to report results across a large range of top-N rather than only N=100. See https://arxiv.org/abs/2411.11767v2 which incorporates a related evaluation protocol and shows how reranker quality can degrade as more documents are reranked.

W3. The masking approach in sec 4.1 seems very similar to margin-based methods, yet no margin-based methods are explored. It would be interesting to see if contrastive loss with additive margin yields a similar effect. https://arxiv.org/abs/2203.02167

W4. There are more losses that can be explored. The RD-Suite paper https://arxiv.org/abs/2306.04455 is focused on distillation, but it is a good starting point for exploring other losses. RankGPT and Pairwise Rank Prompting (PRP) are not well suited for multimodal reranking unless multiple documents can be represented in the same context.

W5. Importantly, not many approaches for negative selection are explored. Given that one of the main weaknesses of CL is small gradients, likely caused by easy negatives, it seems important to explore the proposed framework under different negative selection strategies. Previous work shows more than a 10% difference in nDCG is possible when using different negative selection strategies for embedding models, and it's plausible this would apply to rerankers, see https://arxiv.org/abs/2407.15831

W6. Finally, the authors should make it clear whether they plan to release the model or data. These would greatly strengthen the contribution of the paper.

**Questions:**

Did you investigate whether SFT or CL is more robust against hard negatives? Jacob et al investigate reranker failures and find that reranker can get progressively worse as they need to rerank more documents at test time, suggesting they are undertrained against noisy negatives. See: https://arxiv.org/abs/2411.11767

---

> ### Author Response · Authors · 2025-11-21
> **Reply to Weakness 1-3**
>
> We sincerely thank your insightful comments and address the concerns accordingly.
>
> ---
>
> # Reply to W1:
> > The paper is not self contained in the main text. Many of the main results are in the appendix. For this statement "Rival and surpass leading textual reranker", there is no easy way for the reader to verify. The relevant results are in the appendix, and not averaged across datasets.
>
> We sincerely thank you for pointing this out. Due to space limitations in the main text, we presented the detailed per-dataset results in Appendix E.1, while the main table reports the average performance across categories and the overall mean results. We acknowledge that the appendix currently does not include category-level and overall averages, which may make verification less convenient for readers. We apologize for this oversight and will include these averaged results in the revised version to facilitate easier comparison and validation.
>
> ---
>
> # Reply to W2:
> > There are only experiments when reranking the top-100 retrieved documents, but many real world settings will require reranking more or less than this amount. Given that the reranker here is pointwise, it would be relatively straightforward to report results across a large range of top-N rather than only N=100. See https://arxiv.org/abs/2411.11767v2 which incorporates a related evaluation protocol and shows how reranker quality can degrade as more documents are reranked.
>
> Thank you for highlighting this point. We agree that real-world applications often require reranking varying numbers of retrieved documents, and that evaluating only at top-100 may mask important behaviors. In fact, during our preliminary investigations with Jina-Rerank-m0 and MonoQwen2-VL-v0.1, we also observed similar phenomena—namely, that reranking quality can degrade as the number of candidates increases. This reinforced our decision to follow the evaluation protocol of [Qwen3-Reranker](https://arxiv.org/abs/2506.05176)[1], which adopts top-100 as a standardized and comparable setting across models.
>
> To further address your concern, we additionally evaluate our model and strong baselines ([Qwen3-Reranker](https://arxiv.org/abs/2506.05176)[1] and [Jina-Rerank-m0](https://huggingface.co/jinaai/jina-reranker-m0)) under both top-25 and top-100 settings. The results are shown in the table. For text-to-text retrieval, all models improve when moving from top-25 to top-100, and the magnitude of improvement is consistent with their relative ranking—stronger models benefit more from the larger candidate pool. On the full MRB benchmark, our models (GMR-3B / GMR-7B) also show consistent gains (+1.47% / +2.39%), whereas Jina-Rerank-m0 exhibits a slight decrease (-0.08%). These observations suggest that our model maintains robust performance across different top-N configurations.
>
> |  | T2T  |   |   |   | ALL |   |   |
> |-----|:---:|:---:|:---:|:---:|:-----:|:---:|:---:|
> |Model|Top25|Top100|Δ|-|Top25|Top100|Δ|
> |Qwen3-Reranker|58.87|60.49|+1.63|-|-|-|-|
> |Jina-Rerank-m0|54.93|55.36|+0.43|-|54.44|54.36|-0.08|
> |GMR-3B|58.03|59.22|+1.19|-|59.93|61.40|+1.47|
> |GMR-7B|59.15|61.08|+1.93|-|61.46|63.85|+2.39|
>
> We will include these additional results in the revised version to provide a more complete picture of the model’s behavior under varying reranking depths.
>
> [1] Qwen3 Embedding: Advancing Text Embedding and Reranking Through Foundation Models, Arxiv, 2025
>
> ---
>
> # Reply to W3：
>
> > The masking approach in sec 4.1 seems very similar to margin-based methods, yet no margin-based methods are explored. It would be interesting to see if contrastive loss with additive margin yields a similar effect. https://arxiv.org/abs/2203.02167
>
> Thank you for the thoughtful observation. The masking approach indeed resembles margin-based techniques, which adjust positive-example scores through an additive margin to mitigate issues in contrastive learning. However, in the reranking setting, the score scales and embedding–model characteristics differ substantially from those in typical embedding-based retrieval tasks. As a result, applying margin-based hyperparameters directly may not transfer well and likely requires careful re-examination and adaptation.
>
> We are currently seeking additional computational resources to explore this direction, and we plan to investigate whether margin-based contrastive formulations can be effectively integrated into reranking models in future work.
>
> ---

---

> ### Author Response · Authors · 2025-11-21
> **Reply to Weakness 4-6, Question 1**
>
> # Reply to W4
>
> > There are more losses that can be explored. The RD-Suite paper https://arxiv.org/abs/2306.04455 is focused on distillation, but it is a good starting point for exploring other losses. RankGPT and Pairwise Rank Prompting (PRP) are not well suited for multimodal reranking unless multiple documents can be represented in the same context.
>
> Thank you for pointing out the broader landscape of loss functions. We agree that the multimodal reranking setting poses unique challenges compared to text-only rerankers. In particular, multimodal documents often contain longer and more complex representations (e.g., high-resolution images), making it difficult to place multiple candidates in a single context window. This limitation naturally reduces the applicability of pairwise or listwise methods such as RankGPT or PRP in current multimodal settings.
>
> Regarding distillation-based approaches such as those explored in RD-Suite, we note that progress in this direction has been limited partly because multimodal reranking currently lacks a strong, high-quality teacher model, and because the cost of training distillation pipelines is significantly higher than in text-only scenarios. That said, we believe our trained models—especially GMR-7B—can serve as strong teacher models in future work. This opens up promising opportunities for distillation-based multimodal reranking methods and for exploring richer loss functions beyond SFT and CL.
>
> ___
>
> # Reply to W5
>
> > Importantly, not many approaches for negative selection are explored. Given that one of the main weaknesses of CL is small gradients, likely caused by easy negatives, it seems important to explore the proposed framework under different negative selection strategies. Previous work shows more than a 10% difference in nDCG is possible when using different negative selection strategies for embedding models, and it's plausible this would apply to rerankers, see https://arxiv.org/abs/2407.15831
>
> For negative sampling, we follow prior work and adopt a 1:1 ratio of random and hard negatives during training, as suggested in prior studies (e.g., [mGTE](https://arxiv.org/abs/2407.19669) ) on negative sampling in reranking. For hard negative mining, we follow strategies commonly used in multimodal embedding models (e.g., [GME](https://arxiv.org/abs/2412.16855) ), which directly extend to our reranking setting.
>
> To verify our sampling strategy, we additionally conduct experiments on a pure text retrieval task (MS MARCO), using the same SFT/CL training settings as in the main paper, and evaluate in the T$\rightarrow$T test datasets. We compare three strategies: Random-only, Balanced (1:1), and Hard-only negative sampling. The results are shown below:
>
> |Model|	Random	| Balance |	Hard |
> |-----|:---------:|:---------:|:------:|
> |SFT  | 51.91	| 52.63	  |52.55 |
> |CL	  | 48.96   | 51.55	  |50.58 |
>
> The balanced 1:1 sampling strategy consistently provides the best performance, further supporting its effectiveness and generalizability.
>
> [1] mGTE: Generalized Long-Context Text Representation and Reranking Models for Multilingual Text Retrieval, EMNLP, 2024
>
> [2] GME: Improving Universal Multimodal Retrieval by Multimodal LLMs, CVPR, 2025
>
> ___
>
> # Reply to W6
>
> > Finally, the authors should make it clear whether they plan to release the model or data. These would greatly strengthen the contribution of the paper.
>
> All the datasets used in our work are derived from publicly available sources (e.g., [M-BEIR](https://huggingface.co/datasets/TIGER-Lab/M-BEIR), [Vidore](https://arxiv.org/abs/2407.01449); refer to Section 5.1). In addition, we plan to release our trained model to further support and encourage research on general multimodal reranking.
>
> [1] UniIR: Training and Benchmarking Universal Multimodal Information Retrievers, ECVA, 2024
>
> [2] ColPali: Efficient Document Retrieval with Vision Language Models, ICLR, 2025
>
> ---
>
> # Reply to Q1
>
> > Did you investigate whether SFT or CL is more robust against hard negatives? Jacob et al investigate reranker failures and find that reranker can get progressively worse as they need to rerank more documents at test time, suggesting they are undertrained against noisy negatives. See: https://arxiv.org/abs/2411.11767
>
> Thank you for the insightful question. To examine robustness under harder negative conditions, we expand our evaluation from Top-25 to Top-100 reranking. Both SFT and CL show performance improvements when increasing the number of documents to rerank, indicating that our training is effective even under noisier negative distributions.
>
> Importantly, SFT exhibits a larger performance gain, suggesting that it is more robust to hard negatives compared to CL.
>
> | Model | Top25 |	Top100 |  $\Delta$ |
> |-------|:-------:|:----------:|:-----------:|
> | SFT   | 58.09 |	58.88  |	+0.79  |
> | CL    | 56.40 |	56.76  |	+0.36  |

---

> ### Author Response · Authors · 2025-11-26
>
> Dear Reviewer 51SZ：
>
> We have revised our paper and highlighted all modifications in blue. Specifically, in Section 5.2, we add a comparative analysis of model performance under different top-K settings; in Appendix E.1, we report the average score of each model on each dataset category as well as the overall averages; and in Appendix E.4, we include experiments and analyses on negative-sample selection strategies in the text-only setting. We hope these additional experiments address your concerns. We sincerely appreciate your insightful feedback and look forward to your response and further discussion.

---

### Author Response · Authors · 2025-12-03

Dear Area Chairs,

Below is a consolidated summary of our rebuttal:

1. **Clarifying reranking performance in the standard text-only setting** (Reviewers XjCJ and W9sD). To address concerns regarding model performance in the standard textual reranking scenario, we repeat the experiments in a text-only setting and obtain conclusions consistent with those from the multimodal scenario—SFT consistently outperforms CL. This demonstrates that our findings generalize robustly to traditional text-only reranking as well. (See the reply to W2 of XjCJ and Q3 of W9sD.)
2. **Addressing concerns regarding performance degradation as the top-K size increases** (Reviewers 51SZ and W9sD). We conduct comparative evaluations under top-25 and top-100 settings. First, both SFT and CL show improved performance when moving from top-25 to top-100, confirming the robustness of our approach, with SFT exhibiting consistently larger gains, suggesting stronger resilience in scenarios with more negative candidates. Second, our final models (GMR-3B / GMR-7B) also demonstrate stable improvements from top-25 to top-100 and greater robustness relative to Jina-rerank-m0. (See the reply to W2 of 51SZ and W2 of W9sD.)
3. **Analyzing performance under different negative-sample selection strategies** (Reviewers 51SZ, XjCJ, and xxqB). We conduct experiments under three strategies—fully random negatives, a 1:1 mixture of random and hard negatives, and fully hard negatives. Results show that the mixed strategy achieves the best average performance for both SFT and CL, aligning with prior work and validating our sampling design. Moreover, SFT exhibits stronger robustness across sampling strategies than CL, further highlighting its methodological advantage. (See the reply to W5 of 51SZ, Q2 of XjCJ, and Q2 of xxqB.)
4. **Improving clarity and addressing writing issues noted in the reviews** (Reviewers 51SZ and W9sD). Following the reviewers’ suggestions, we revise sections with unclear descriptions and mark all changes in blue. In particular, we refine the exposition of our method in Section 3 and add average score in Appendix E.1 to improve readability. (See reply to W1 of 51SZ and W3/W4 of W9sD.)
5. **Addressing concerns about potential overfitting to the MRB benchmark and model behavior in multi-image scenarios** (Reviewer xxqB). We clarify the generalization behavior of our model and add new experiments on the Knowledge subset of MRMR in multi-image settings. The results show that our model maintains strong and stable performance even with multiple input images, demonstrating its robustness and validating the effectiveness of our approach. (See reply to W2 and Q1 of xxqB.)

In summary, we have addressed the majority of the reviewers’ concerns and resolved all reasonable issues raised during the review process, providing additional analyses where necessary. We sincerely thank all reviewers and the Area Chairs for their careful evaluation and constructive feedback. All authors are deeply grateful for your time and consideration.

---

### Meta-Review · Area_Chair_B2v1 · 2026-01-06

**Summary:**

The paper studies the role of the training loss for LLM-based re-ranking. It is argued that standard SFT can be superior to traditional contrastive learning (CL) objectives. This is demonstrated empirically, and then analyzed in terms of the weight and direction of the updates for each. Based on these, the authors establish that SFT based re-rankers can achieve SoTA performance on an image-text benchmark.

Reviewers were generally positive about the work, with the following being the main critiques:
- **Significance and scope of technical results**. Multiple reviewers noted that the main technical novelty is limited, and/or not especially surprising. Concretely, the central observation appears to be that the SFT loss involves decoupled terms for each negative, while the CL loss involves a single joint softmax over all negatives, which causes a difference in the influence on learned weights.
- **Limited exploration of design space**. Multiple reviewers noted that the scope of analysis is a little narrow, with treatment of listwise rankers, negative sampling, and the loss function having more design choices that could affect the results.
- **Restriction to image-text retrieval**. Multiple reviewers noted the restriction to a particular image-text retrieval benchmark as being potentially limiting, noting that results in text retrieval would be more standard.
- **Role of negative size**. Multiple reviewers suggested it was unclear how the size of the negative set (in training and inference) affected the findings.
- **Issues with clarity**. Multiple reviewers that while the overall message of the paper was clear, there were issues in presentation in certain parts, and that certain details were not fully clear from reading the body alone.

**Reviewer Concerns:**

- **Significance and scope of technical results**. The authors acknowledged there could be further scope for theoretical analysis, including the study of listwise losses and a more precise characterization of why decoupled losses can outperform coupled ones. However, they emphasized the significance of the finding that SFT losses can outperform CL.
  - *Partially addressed*. The authors do not seem to dispute that the scope of the results is as understood by the reviewers; the argument appears to simply be that the findings are by themselves of sufficient interest to the community. From our reading, we must agree with Reviewer XjCJ that the analysis in Section 3.3 is a little shallow, and the authors may consider toning down claims of providing a theoretical explanation for the difference between SFT and CL. Nonetheless, we do agree that the finding of SFT's superior performance is of broad interest.
- **Limited exploration of design space**. The authors acknowledged that these design choices could be of interest for future work. They did argue that the choices in the present paper are standard practice. For the issue of negative sampling, the authors presented some results showing that their balanced sampling approach leads to better results that sampling only hard negatives.
  - *Partially addressed*. These issues remain largely open, although the authors' argument that the setting considered is a standard one is reasonable.
- **Restriction to image-text retrieval**. The authors added results in a text-only retrieval setting (MSMARCO), as well as a multi-image retrieval setting (MRMR), both of which showed positive results.
  - *Mostly addressed*. The newly added results further expand the scope of the paper, and strengthen the main claims.
- **Role of negative size**. The authors argued that for training, it is common to use a relatively small number of negatives ($1-16$), owing to the large batch sizes. The authors presented additional results for inference, where the number of negative documents increased from $25$ to $100$. Here, they found that with more negatives, the gap between SFT and CL increases.
  - *Mostly addressed*. The new inference results are interesting, and are consistent with the paper's central claims.
- **Issues with clarity**. The authors promised to update the Appendix and body to make the paper more self-contained, and to fix typos.
  - *Partially addressed*. These changes do not seem to have been fully made in the latest revision. From our reading, we wish to add that the mathematical formalism could be improved.
    - It is strongly advised not to use the multi-character "ins" as-is; rather, consider just "i", or perhaps $\sf{ins}$.
    - The use of $\sigma$ is generally understood to correspond to a sigmoid, while here it refers to a score; consider using $f$ instead.
    - Equation 4 is unnecessarily verbose and apparently diverges from Equation 1.
    - The loss after Equation 7 should appear at the outset, before mention is made of derivatives.
    - A precise mathematical specification of Algorithm 1 is lacking, which is a surprising omission.

**Reviewer Scores:**

- **W9sD**: the reviewer noted they wished to keep their original score of 6.
- **XjCJ**: the reviewer noted they wished to keep their original score of 4.
- **51SZ**: as the original review was fairly positive, without qualifications placed on the score, we think it likely to have remained at 8.
- **xxqB**: as the reviewer's concern about the focus on image-text benchmarks was partly addressed with the new MRMR and MSMARCO results, we think it possible the reviewer could increase their score to 8.

---

### Decision · Program_Chairs · 2026-01-26

Accept (Poster)